# Learning Explicit Single-Cell Dynamics Using ODE Representations

**Jan-Philipp von Bassewitz**[1,2]**, Adeel Pervez**[1]**, Marco Fumero**[1]**,**
**Matthew Robinson**[1]**, Theofanis Karaletsos**[2]**, Francesco Locatello**[1,2]

[1]Institute of Science and Technology Austria (ISTA),
[2]Chan Zuckerberg Initiative (CZI)

## Abstract

Modeling the dynamics of cellular differentiation is fundamental to advancing the understanding and treatment of diseases associated with this process, such as cancer. With the rapid growth of single-cell datasets, this has also become a particularly promising and active domain for machine learning. Current state-of-the-art dynamics models, however, rely on computationally expensive optimal transport preprocessing and multi-stage training, while also not directly learning explicit gene interactions. To address these challenges, we propose Cell-Mechanistic Neural Networks (*Cell-MNN*), an encoder-decoder architecture whose latent representation is a *locally linearized ODE* governing the dynamics of cellular evolution from stem to tissue cells. Cell-MNN is fully end-to-end (besides a standard PCA pre-processing) and its ODE representation learns interpretable gene interactions. Empirically, we show that Cell-MNN achieves competitive performance on single-cell benchmarks, surpasses state-of-the-art baselines in scaling to larger datasets and joint training across multiple datasets, while also learning interpretable gene interactions that we validate against the TRRUST database.

github.com/czi-ai/cell-mnn

## 1 Introduction

The process by which stem cells differentiate into specialized tissue cells is poorly understood, and prediction of cellular fate remains an open problem in systems biology. Deeper understanding of the differentiation dynamics is essential for advancing treatment of diseases such as cancer (Chu et al., 2024), neurodegenerative diseases (Cuomo et al., 2023), and to improving wound healing (Rodrigues et al., 2019). While all cells in an organism share the same genome, the level of expression of genes varies over time as differentiation progresses. During this process, genes activate or repress the expression of other genes through complex regulatory mechanisms, causing the cell to differentiate.

Today, only a small subset out of the large number of possible gene interactions has been thoroughly studied. This is due to both the vast combinatorial search space, with $\sim 10^8$ theoretically possible gene interactions, and the experimental effort required to validate specific mechanisms. However, recent advances in single-cell sequencing technology (Macosko et al., 2015; Zheng et al., 2017) have enabled high-throughput measurements that were previously prohibitively expensive, producing datasets that are growing at a pace exceeding Moore's law (Kharchenko, 2021). This rapid growth, coupled with the limitations of direct experimental approaches, presents a unique opportunity to apply machine learning methods to study single-cell dynamics.

In this work we propose Cell-MNN, a method to jointly tackle the challenges of predicting cell fate and discovering gene regulatory interactions. Cell-MNN is an end-to-end encoder-decoder architecture whose representation is a *locally linear* ordinary differential equation (ODE) that governs the dynamics of cellular evolution from stem to tissue cells. The ODE representation of Cell-MNN learns explicit and interpretable gene interactions.

A key challenge in modeling single-cell dynamics is that cells are destroyed by measurement, resulting in datasets that contain a single point along each cell's trajectory (Tong et al., 2020), i.e., a

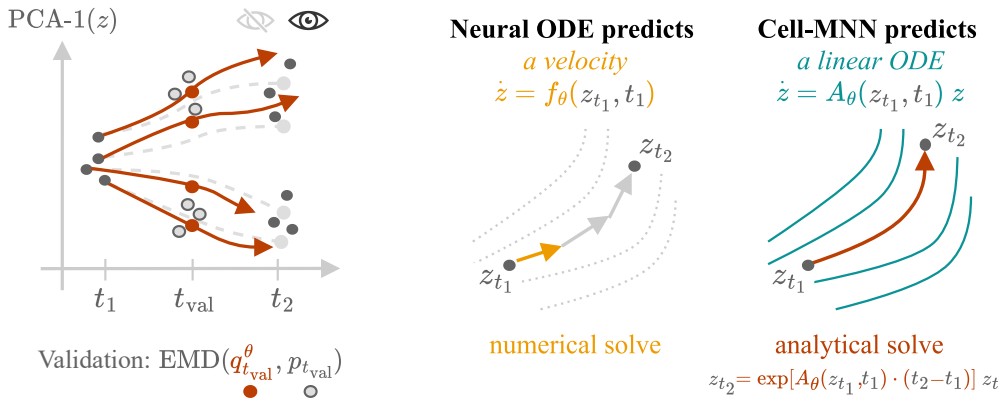

(a) *Single-cell interpolation problem*    (b) *Difference of Cell-MNN with respect to NODEs*

Figure 1: **(a)** Single-cell interpolation: trajectories are evaluated by the earth mover's distance (EMD) between predictions and the marginal distribution at a held-out time $t_{val}$. **(b)** Like a hypernetwork, Cell-MNN predicts a linear operator $\boldsymbol{A}_\theta(\boldsymbol{z}, t)$ that approximates the local dynamics explicitly, whereas Neural ODEs and Flow Matching models only output a velocity.

*snapshot observation.* This motivated a line of work on reconstructing trajectories from snapshot data: The best-performing methods in this setting rely on optimal transport (OT) preprocessing to create label trajectories (Tong et al., 2024a; Zhang et al., 2025; Kapusniak et al., 2024; Wang et al., 2025), which becomes a computational bottleneck for large datasets due to quadratic scaling of the Sinkhorn algorithm with the number of samples (Cuturi, 2013). In contrast, Cell-MNN eliminates OT preprocessing entirely and is designed to be end-to-end. Another bottleneck of state-of-the-art (SOTA) models such as OT-MFM (Kapusniak et al., 2024) and DeepRUOT (Zhang et al., 2025) is that they involve multiple training stages and networks, making amortized training across datasets challenging, whereas Cell-MNN is trained in a single stage, enabling straightforward amortized training across multiple datasets. Furthermore, existing SOTA methods focus on accurate interpolation of empirical distributions and do not directly learn explicit gene interactions. By comparison, Cell-MNN's ODE representation exposes the gene interactions used to predict the cellular evolution. While there are dedicated methods for discovering gene interactions (Lin et al., 2025), to the best of our knowledge, no such method achieves SOTA predictive performance on single-cell interpolation benchmarks. Cell-MNN addresses both challenges simultaneously, bridging the gap between predictive performance and interpretable gene regulatory modeling.

**Contributions.** Our main contributions are: **(i)** we propose Cell-MNN, an architecture that models single-cell dynamics via a locally linearized ODE representation; **(ii)** we demonstrate SOTA average performance on three benchmark datasets; **(iii)** we show that eliminating OT preprocessing enables scalability, with Cell-MNN outperforming all baselines on upsampled datasets; **(iv)** we leverage the end-to-end design for amortized training across datasets, surpassing a strong amortized baseline; and **(v)** we exploit the explicit ODE representation to extract gene interactions and quantitatively validate them against the TRRUST database (Han et al., 2018) of gene interactions.

## 2 LEARNING THE DYNAMICS OF CELLS

**Formalizing the Problem.** We assume a data-generating process consisting of a cell state $\boldsymbol{c}(t) \in \mathcal{C}$ evolving over time in a high-dimensional state space $\mathcal{C}$ that includes all relevant molecular, physical, and biochemical variables, and an observation function mapping this state to data. The measurement process observes only a subset of the full state mapping it to the *gene expression vector* of $d_x$ genes $\boldsymbol{x}_t \in \mathbb{R}^{d_x}$ via an unknown, potentially noisy measurement process $\boldsymbol{m} : \mathcal{C} \to \mathbb{R}^{d_x}$, so that $\boldsymbol{x}(t) = \boldsymbol{m}(\boldsymbol{c}(t))$. Measuring the system involves deconstructing the observed cell, which implies that each measurement corresponds to a single point along its trajectory, i.e., a *snapshot observation*. We assume time $t \in \mathbb{R}$ to be a continuous variable and denote an arbitrary time interval by $\Delta t \in \mathbb{R}$. In practice, the lab schedules a discrete set of experimental time points $\mathcal{T} = \{t_1, t_2, \dots, t_K\}$ at which cell populations are sampled. We denote by $p_t$ the distribution of $\boldsymbol{x}_t$ at time $t$. The dataset of snap-

shot observations is $\mathcal{D} = \{\boldsymbol{x}^{(i)}, t^{(i)}\}_{i=1}^{N}$ with $t^{(i)} \in \mathcal{T}$, and our goal is to learn a best-fit mechanistic model for the dynamics of the observable $\boldsymbol{x}_t$ that is consistent with the family of marginals $\{p_t\}_{t \in \mathcal{T}}$.

## 2.1 CELL-MNN

SOTA models on single-cell interpolation benchmarks face scalability issues from OT preprocessing and do not learn interpretable gene interactions that can be cross-validated against biological evidence. Our goal is to design a scalable mechanistic model of single-cell dynamics using an ODE representation, enabling accurate forecasting and discovery of interpretable gene interactions.

The Mechanistic Neural Network (MNN) is a recent architecture that Pervez et al. (2024) showed to outperform Neural ODEs on tasks such as solar system dynamics and the $n$-body problem, while also being able to learn explicit models of the underlying dynamics. This motivates us to design an MNN-inspired architecture for the single-cell setting. However, this domain presents unique challenges that make the vanilla MNN not directly applicable: for ODE discovery, the MNN has only been applied with full trajectories and not yet in biological contexts. Moreover, when identifying a latent space ODE with the MNN, there is typically no way to interpret that ODE in the input space. In contrast, single-cell dynamics require learning latent space dynamics from *snapshot data*. To discover gene interactions, the learned ODE must furthermore be interpretable in the input space. We therefore adapt the MNN architecture to this setting and refer to the resulting version as *Cell-MNN*. Cell-MNN is an encoder-decoder model, learning a mechanistic map

$$\boldsymbol{x}_{t+\Delta t} = \text{Cell-MNN}_\theta(\boldsymbol{x}_t, t, \Delta t),$$

which maps a gene expression vector $\boldsymbol{x}_t$ at time $t$ to a predicted state $\boldsymbol{x}_{t+\Delta t}$ after an *arbitrary* time interval $\Delta t$. We define the model-induced distribution at time $t + \Delta t$ as $q_{t+\Delta t}^\theta$, which is the distribution of Cell-MNN$_\theta(\boldsymbol{x}_t, t, \Delta t)$ when $\boldsymbol{x}_t$ is drawn from $p_t$. As a core part of the architecture, Cell-MNN maps to a compressed representation $\boldsymbol{z} \in \mathbb{R}^{d_z}$, with $d_z \ll d_x$, of the high-dimensional gene expression vector $\boldsymbol{x} \in \mathbb{R}^{d_x}$, and learns the dynamics in the latent space. Following prior work (Tong et al., 2024a), we obtain this latent representation by applying principal component analysis (PCA), with projection matrix $\boldsymbol{V}_{\text{PCA}} \in \mathbb{R}^{d_x \times d_z}$, so that $\boldsymbol{z} = \boldsymbol{V}_{\text{PCA}}^\top \boldsymbol{x}$.

**Locally Linearizing the Latent ODE.** The latent vector $\boldsymbol{z} \in \mathbb{R}^{d_z}$ in the PCA subspace is assumed to follow non-autonomous, non-linear dynamics $\dot{\boldsymbol{z}} = \boldsymbol{f}(\boldsymbol{z}, t)$. In practice, this ODE is often highly complex, and learning an *explicit* form that globally approximates it would be intractable due to the combinatorial search space of basis functions that grows with increasing latent space dimension $d_z$.

To address this, we decompose the intractable global ODE discovery problem into smaller subproblems: at the current state $(\boldsymbol{z}^{(i)}, t^{(i)})$, which we also call the *operating point*, we approximate the dynamics by a linear ODE in a small neighborhood. The learning task is then to predict these local dynamics models from the operating point $(\boldsymbol{z}^{(i)}, t^{(i)})$ using an encoder.

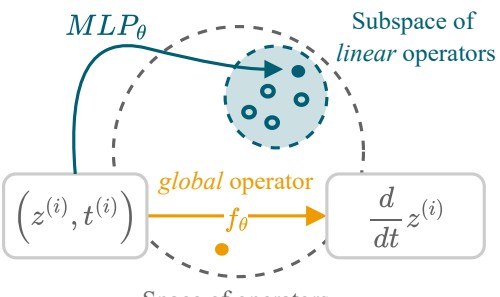

$$\dot{\boldsymbol{z}} = \boldsymbol{f}(\boldsymbol{z}, t)$$
$$= \boldsymbol{A}(\boldsymbol{z}, t)\,\boldsymbol{z}, \qquad \text{if } \boldsymbol{f}(\boldsymbol{0}, t) = \boldsymbol{0},\ \forall t \in \mathbb{R},$$
$$\approx \boldsymbol{A}_\theta\big(\boldsymbol{z}^{(i)}, t^{(i)}\big)\,\boldsymbol{z}.$$

We predict the linear operator $\boldsymbol{A}_\theta \in \mathcal{A}$ using a multilayer perceptron MLP$_\theta$ : $\mathbb{R}^{d_z+1} \to \mathcal{A}$. Here $\mathcal{A} := \mathcal{L}(\mathbb{R}^{d_z}, \mathbb{R}^{d_z}) \cong \mathbb{R}^{d_z \times d_z}$ represents the space of linear operators acting on $\mathbb{R}^{d_z}$. Note that, while the operator governing the local dynamics is linear, it is a non-linear function of the current latent state $\boldsymbol{z}^{(i)}$ and time $t^{(i)}$. In Appendix D, we show that the reparametrization of the right-hand side $\boldsymbol{f}(\boldsymbol{z}, t) = \boldsymbol{A}(\boldsymbol{z}, t)\,\boldsymbol{z}$ always exists under mild assumptions.

Figure 2: Visualization of the meta-learning task of Cell-MNN's encoder: Rather than directly predicting the velocity at a given operating point, as in the Neural ODE framework, the MLP of Cell-MNN maps to the space of linear operators. Conditioned on the current system state, it predicts local linear approximations to the global dynamics.

This approach is conceptually orthogonal to Neural ODEs (Chen et al., 2018), which learn an *unconditional black-box* approximation to $\boldsymbol{f}$. In the Cell-MNN setting, the MLP functions more like a *hypernetwork* (Ha et al., 2017), outputting a conditional white-box linear function $\boldsymbol{g}_\theta(\boldsymbol{z}, t | \boldsymbol{z}^{(i)}, t^{(i)}) = \boldsymbol{A}_\theta(\boldsymbol{z}^{(i)}, t^{(i)}) \boldsymbol{z}$ that locally approximates $\boldsymbol{f}$ at the operating point $(\boldsymbol{z}^{(i)}, t^{(i)})$. Unlike most neural operators (Li et al., 2021; Kovachki et al., 2021) that learn a *single* global operator, Cell-MNN predicts a state-conditioned linear operator for each operating point. This makes the learned dynamics explicit and enables amortization across arbitrarily many states and datasets within a single network.

**Decoding by Analytically Solving the ODE.** Decoding the ODE representation involves solving the ODE system. The locally linearized formulation of the dynamics has the advantage that the latent space ODE admits a local closed-form solution. For fixed $\boldsymbol{A}_\theta$ at the operating point, the system $\dot{\boldsymbol{z}} = \boldsymbol{A}_\theta \boldsymbol{z}$ is a linear, time-invariant ODE with solution

$$\boldsymbol{z}(t^{(i)} + \Delta t) = \exp\left(\boldsymbol{A}_\theta \Delta t\right) \boldsymbol{z}_t^{(i)}.$$

Predictions in the gene expression space are obtained by projecting back $\boldsymbol{x}(t + \Delta t) = \boldsymbol{V}_{\text{PCA}} \boldsymbol{z}(t + \Delta t)$.

**Parametrization of the Operator.** For more fine-grained control over the parametrization of $\boldsymbol{A}_\theta$, we let the MLP predict the matrix in an eigen-decomposed form $\boldsymbol{A}_\theta = \boldsymbol{P}_\theta \operatorname{diag}(\boldsymbol{\lambda}_\theta) \boldsymbol{P}_\theta^{-1}$, which is also beneficial to compute the matrix exponential. To ensure invertibility of $\boldsymbol{P}_\theta$, we train with the additional regularizer $\mathcal{L}^{\text{inv}}(\theta) = 1/(\det(\boldsymbol{P}_\theta) + \epsilon)$, which is practical if the latent space is small. This also lets us introduce inductive bias by selectively fixing eigenvalues, for example to zero, if needed.

**Optimization.** We train the MLP parameters $\theta$ by minimizing the Maximum Mean Discrepancy (MMD, Gretton et al. (2012))[1] between the model-induced marginals $q_t^\theta$ and the empirical marginals $\mu_t$, thereby fitting a mechanistic model whose dynamics align with the target marginals $p_t$ under a future discounting factor $\gamma$. All discrepancies are computed in latent space via the pullback kernel

$$k_x(\boldsymbol{x}, \boldsymbol{x}') := k_z\left(\boldsymbol{V}_{\text{PCA}}^\top \boldsymbol{x}, \ \boldsymbol{V}_{\text{PCA}}^\top \boldsymbol{x}'\right),$$

so that $\text{MMD}^2(q_t^\theta, p_t; k_x) = \text{MMD}^2(q_t^{\theta,z}, p_t^z; k_z)$. Here, $p_t^z$ and $q_t^{\theta,z}$ denote the distributions of the gene expression marginals in the latent space. The MMD loss is:

$$\mathcal{L}^{\text{MMD}^2}(\theta) = \mathbb{E}_t\left[\sum_{t'=t}^{t_K} \gamma^{t'} \text{MMD}^2\left(q_{t'}^\theta, p_{t'}; k_x\right)\right].$$

Following Tong et al. (2020), we also regularize the kinetic energy to improve generalization:

$$\mathcal{L}^{\text{kin}}(\theta) = \mathbb{E}_{t, \boldsymbol{z}_t \sim q_t^\theta}\left[\|\dot{\boldsymbol{z}}_t\|^2\right] = \mathbb{E}_{t, \boldsymbol{z}_t \sim q_t^\theta}\left[\|\boldsymbol{A}_\theta(\boldsymbol{z}_t, t)\, \boldsymbol{z}_t\|^2\right],$$

which serves as a soft constraint encouraging trajectories close to optimal transport flows in the sense of the Benamou & Brenier (2000) formulation. Our final loss then becomes:

$$\mathcal{L}^{\text{total}}(\theta) = \mathcal{L}^{\text{MMD}^2}(\theta) + \lambda_{\text{kin}}\mathcal{L}^{\text{kin}}(\theta) + \lambda_{\text{inv}}\mathcal{L}^{\text{inv}}(\theta). \tag{1}$$

**Computational Complexity.** With $\boldsymbol{A}_\theta$ given in eigendecomposed form at an operating point, evaluating the analytical solution (Eq. 2.1) at $T$ time points has time complexity $\mathcal{O}(T d_z^2)$ and space complexity $\mathcal{O}(d_z^2)$, where $d_z$ is the latent space dimensionality. This improves the time and space complexity over the Scalable Mechanistic Neural Network (S-MNN) (Chen et al., 2025). Forming the full operator requires computing $\boldsymbol{P}_\theta^{-1}$, incurring a one-time $\mathcal{O}(d_z^3)$ cost per operating point.

**Limitations.** The cubic time complexity in the latent dimension can become a challenge for very high-dimensional latent spaces but could be mitigated by imposing sparsity assumptions on $\boldsymbol{A}_\theta$. In our application to single-cell dynamics, we follow the practice (Tong et al., 2020; 2024a) of using a 5, 50 and 100-dimensional PCA space, which captures most of the variance of the data and, crucially, resolves cell-type information (see Appendix E). Note that OT preprocessing on two time points, when using the Sinkhorn algorithm, scales as $\mathcal{O}(d_z n^2)$ with the number of samples $n$, which becomes a bottleneck for large datasets, as $n$ is usually much larger than $d_z$. However, approximate batch approaches are also possible to address this (Tong et al., 2024a). A separate limitation of predicting the local dynamics is that evolving the system too far may cause it to leave the regime where the linear ODE is accurate, which would require a new forward pass through the encoder to update the ODE. In our experiments, however, we did not encounter this issue.

---

[1]A full definition of the MMD is given in Appendix B

**Uncovering Local Gene Regulatory Interactions.** Combining the linear projection to the PCA subspace $z = V_{\mathrm{PCA}}^\top x$ with locally linear dynamics around an operating point $\dot{z} = A_\theta z$ enables projecting the predicted local dynamics back into the gene expression space with:

$$\frac{d}{dt} z = A_\theta z \iff \frac{d}{dt}\big(V_{\mathrm{PCA}}^\top x\big) = A_\theta V_{\mathrm{PCA}}^\top x \iff \frac{d}{dt} x = V_{\mathrm{PCA}} A_\theta V_{\mathrm{PCA}}^\top x.$$

which gives direct access to an *explicit* form of the predicted local dynamics in the gene expression space, essentially uncovering the predicted local gene regulatory interactions. We interpret:

$$w_{j\to i}(\boldsymbol{x}, t) := \big[V_{\mathrm{PCA}} A_\theta(\boldsymbol{x}, t) V_{\mathrm{PCA}}^\top\big]_{i,j} \cdot \boldsymbol{x}_j,$$

as the interaction weight of gene $j$ to gene $i$. It essentially represents the contribution of gene $j$'s expression to the time derivative of $\boldsymbol{x}_i$. This makes our proposed approach fully interpretable, as we can inspect the learned gene interactions directly.

## 3 RELATED WORKS

**Single-cell Interpolation.** The single-cell trajectory inference problem, as formalized by Lavenant et al. (2023), entails reconstructing continuous dynamics from snapshot data. Early work based on recurrent neural networks (Hashimoto et al., 2016) was followed by Neural ODE-based methods (Tong et al., 2020; 2023; Zhang et al., 2023; Koshizuka & Sato, 2023; Huguet et al., 2022), in which a neural network outputs the velocity field governing the dynamics. In contrast, Cell-MNN predicts an explicit local dynamics model, which not only facilitates the learning of gene interactions but also circumvents the need for numerical ODE solvers.

A separate line of work avoids simulation by relying on OT preprocessing to approximate cell trajectories (Schiebinger et al., 2019; Bunne et al., 2021), which were also used to train flow-matching models such as by Tong et al. (2024a); Kapusniak et al. (2024); Zhang et al. (2025); Wang et al. (2025); Terpin et al. (2024). However, solving the OT coupling with the Sinkhorn algorithm scales quadratically in the number of samples, creating a major bottleneck for large datasets, which is why Tong et al. (2024a) proposed batch-wise approximation. To address this scalability bottleneck, Cell-MNN is designed to eliminate OT preprocessing entirely. Furthermore, SOTA OT-based models such as OT-MFM and DeepRUOT rely on multiple training stages beyond a standard PCA dimensionality reduction, which complicates amortized training across datasets. In contrast, Cell-MNN involves only a single training stage while achieving competitive performance on single-cell benchmarks. Finally, Action Matching (Neklyudov et al., 2023) also avoids OT preprocessing, but unlike Cell-MNN, it does not learn an explicit form of the underlying dynamics.

**Gene Regulatory Network Discovery.** A complementary line of work assumes that the interactions governing cell differentiation can be represented as a graph, known as a *gene regulatory network* (GRN) (Davidson et al., 2002). Tong et al. (2024b) demonstrated that such GRNs can to some extent be recovered from flow-matching models in the setting of low-dimensional synthetic data as simulated by Pratapa et al. (2020). In contrast, we show that Cell-MNN learns biologically plausible gene interactions directly from *real* single-cell data, validating them against the literature-curated TRRUST database. Additional approaches for GRN discovery include tree-based methods (Huynh-Thu et al., 2010; Moerman et al., 2018), information-theoretic approaches (Chan et al., 2017), regression-based time-series models (Lu et al., 2021), Gaussian processes (Äijö & Lähdesmäki, 2009) and ODE-based models such as PerturbODE (Lin et al., 2025) and SCODE (Matsumoto et al., 2017). However, unlike Cell-MNN, these methods typically learn a static GRN and they are either inapplicable to single-cell interpolation benchmarks or do not deliver competitive performance.

Orthogonal to these static GRN approaches, recent methods for *time-resolved* GRN discovery such as Dynamo (Qiu et al., 2022), SCENIC+ (Bravo González-Blas et al., 2023), Marlene (Hasanaj et al., 2025), and Dictys (Wang et al., 2023) infer time-varying GRNs from RNA-velocity or paired scRNA-seq and scATAC-seq data. In contrast, Cell-MNN operates directly on standard scRNA-seq UMI counts from single-cell interpolation benchmarks and yields context-dependent signed interaction weights as a by-product of fitting the dynamics.

**ODE Discovery.** The idea to learn an explicit ODE representation of the cell differentiation dynamics as pursued by PerturbODE (Lin et al., 2025) and Cell-MNN relates directly to the broader problem of ODE discovery. A seminal method in this area is SINDy (Brunton et al., 2016), which

infers governing equations from data but requires access to full trajectories, making it unsuitable for the snapshot-based single-cell setting. Similar limitations apply to more recent approaches such as MNN and related methods such as ODEFormer (Pervez et al., 2024; Chen et al., 2025; Yao et al., 2024; d'Ascoli et al., 2024), which extend ODE discovery to amortized settings by using neural networks to predict the underlying dynamics from observed trajectories. In contrast, Cell-MNN is qualitatively distinct in learning dynamics from population data. It furthermore learns them in *locally linear* form, an idea with strong precedents in physics and control theory such as the Apollo navigation filter (Schmidt, 1966), the control of a 2-link 6-muscle arm model (Li & Todorov, 2004; Todorov & Li, 2005) and rocket landing (Szmuk et al., 2020). The locally linear parameterization theoretically imposes learning control-oriented structure and, in principle, supports the design of performant controllers as described by (Richards et al., 2023), which could enable design of gene perturbations.

## 4 EXPERIMENTS

In the following, we present four experiments to evaluate Cell-MNN in terms of predictive accuracy, suitability for amortized training, scalability, and assessment of the predicted gene interactions.

**Datasets.** For our experiments, we use 3 commonly studied real single-cell datasets. Following Tong et al. (2020), we include the Embryoid Body (*EB*) dataset from Moon et al. (2019), which after preprocessing contains $\sim 16$ K human embryoid cells measured at five time points over 25 days. For EB, we model the time grid with $\mathcal{T} = \{0, 1, ...4\}$. We also use the CITE-seq (*Cite*) and Multiome (*Multi*) datasets from Burkhardt et al. (2022), as repurposed by Tong et al. (2024a). Both consist of gene expression measurements at four time points of cells developing over seven days, with Cite containing $\sim 31$ K cells and Multi $\sim 33$ K cells after preprocessing. Here we model the time grid with the days of measurement, namely $\mathcal{T} = \{0, 1, 2, 3, 7\}$. We use the datasets as preprocessed by Tong et al. (2020; 2024a), which involves filtering for outliers and normalizing the data.

**Training.** We use the same hyperparameters for all experiments unless stated otherwise. Following Tong et al. (2020), we project gene expression to PCA subspace before training. The MLP used to parameterize $A_\theta$ has depth 4, width 96, leaky ReLU activations, and Kaiming normal initialization (He et al., 2015). For stability, we scale the MLP's last layer by 0.01 at initialization so that predictions of $A_\theta$ start near zero. For the MMD, we use the Laplacian kernel $k(z, z') = \exp[-\frac{\max(||z - z'||_1, \epsilon)}{\sigma \cdot d_z}]$ with parameters $\sigma = 1$ and $\epsilon = 10^{-8}$. We optimize the final loss (Eq. 1) with a batch size per time point of 200, future discount factor $\gamma = 0.1$, initialization scale 0.01, and regularization weights $\lambda_{\text{kin}} = 0.1$ and $\lambda_{\text{inv}} = 1$. Optimization is performed using AdamW (Kingma & Ba, 2017; Loshchilov & Hutter, 2019) with a learning rate of $2 \times 10^{-4}$ and weight decay $1 \times 10^{-5}$. Hyperparameters are selected according to grid search and all experiments are run with three random seeds. We validate every 10 steps, with a patience of 40 validation checks and a maximum training time of 200 minutes. All training runs are performed with one NVIDIA GeForce RTX 2080 Ti per model (11 GB of RAM).

### 4.1 SINGLE CELL INTERPOLATION

Following Tong et al. (2020; 2024a), we evaluate model performance by measuring how closely it reproduces the marginal distribution of a held-out time point. Each intermediate day is left out in cross-validation fashion to obtain one comprehensive score per dataset.

**Metric.** For easy comparison with SOTA methods, we follow Tong et al. (2020) and report results in terms of the 1-Wasserstein distance in the PCA subspace ($W_1$ or otherwise EMD). We use the exact linear programming EMD from the POT (Python Optimal Transport) package (Flamary et al., 2021). The EMD metric measures the minimum cost of transporting probability mass to transform one distribution into another, where a lower score represents a closer match of distributions.

**Baselines.** We compare with the 3 SOTA methods for this task, namely OT-MFM (Kapusniak et al., 2024), OT-CFM (Tong et al., 2024a) and DeepRUOT (Zhang et al., 2025). We found the pre-processing of DeepRUOT to be different from the other approaches, which is why we reran the experiments for DeepRUOT with exactly the same input data as the other methods. We also report the performance of other relevant previous works on this problem as indicated in the results Table 1.

Table 1: Model comparison for single-cell interpolation across the *Cite*, *EB*, and *Multi* datasets, sorted by best average performance. We report the mean ± standard deviation of the EMD metric, along with the average across datasets. Standard deviation is computed over left-out time points. Lower values indicate better performance. Values marked * are computed by us.

| Method | Cite (5D) | EB (5D) | Multi (5D) | Average ↓ |
|---|---|---|---|---|
| TrajectoryNet (Tong et al., 2020) | – | 0.848 | – | – |
| WLF-UOT (Neklyudov et al., 2024) | – | 0.800 ±0.002 | – | – |
| NLSB (Koshizuka & Sato, 2023) | – | 0.777 ±0.021 | – | – |
| SB-CFM (Tong et al., 2024a) | 1.067 ±0.107 | 1.221 ±0.380 | 1.129 ±0.363 | 1.139 |
| [SF]$^2$M-Sink (Tong et al., 2024b) | 1.054 ±0.087 | 1.198 ±0.342 | 1.098 ±0.308 | 1.117 |
| [SF]$^2$M-Geo (Tong et al., 2024b) | 1.017 ±0.104 | 0.879 ±0.148 | 1.255 ±0.179 | 1.050 |
| I-CFM (Tong et al., 2024a) | 0.965 ±0.111 | 0,872 ±0.087 | 1.085 ±0.099 | 0.974 |
| DSB (De Bortoli et al., 2021) | 0.965 ±0.111 | 0.862 ±0.023 | 1.079 ±0.117 | 0.969 |
| I-MFM (Kapusniak et al., 2024) | 0.916 ±0.124 | 0.822 ±0.042 | 1.053 ±0.095 | 0.930 |
| [SF]$^2$M-Exact (Tong et al., 2024b) | 0.920 ±0.049 | 0.793 ±0.066 | 0.933 ±0.054 | 0.882 |
| OT-CFM (Tong et al., 2024a) | 0.882 ±0.058 | 0.790 ±0.068 | 0.937 ±0.054 | 0.870 |
| DeepRUOT (Zhang et al., 2025)* | 0.845 ±0.167 | 0.776 ±0.079 | 0.919 ±0.090 | 0.846 |
| OT-Interpolate* | 0.821 ±0.004 | 0.749 ±0.019 | 0.830 ±0.053 | 0.800 |
| OT-MFM (Kapusniak et al., 2024) | **0.724 ±0.070** | 0.713 ±0.039 | 0.890 ±0.123 | 0.776 |
| Cell-MNN (ours)* | 0.791 ±0.022 | **0.690 ±0.073** | **0.742 ±0.100** | **0.741** |

As an intuitive bar to cross, we additionally compute the performance of solely interpolating the optimal transport map between two consecutive time points and refer to it as OT-Interpolate.

**Results.** Table 1 summarizes the results in 5-dimensional PCA subspace on all three datasets. This setting is the most realistic in terms of scientific application, as trajectory inference is typically done in low-dimensional PCA subspaces, which capture most of the variance (See Appendix E for more details). Cell-MNN achieves the best performance on EB and Multi, and ranks second on Cite, leading to the highest average performance across datasets. Notably, Cell-MNN is the only method that outperforms our proposed OT-Interpolate benchmark on all datasets. We think this is an important additional result, because any method that trains on velocities that are derived from the OT map implicitly treats OT-Interpolate as the ground truth. This also explains the strong performance of OT-Interpolate. In Appendix G.1, we provide additional evaluations in 50- and 100-dimensional PCA subspace showing that Cell-MNN performs competitively in these settings as well.

## 4.2 AMORTIZED TRAINING

| Model | Cite (Inflated) | EB (Inflated) | Multi (Inflated) |
|---|---|---|---|
| I-CFM | 0.0390 ±0.0249 | 0.0403 ±0.0045 | 0.0482 ±0.0144 |
| OT-CFM | | -- OOM Error -- | |
| DeepRUOT | | -- OOM Error -- | |
| Batch-OT-CFM | 0.0232 ±0.0041 | 0.0243 ±0.0025 | 0.0302 ±0.0010 |
| Cell-MNN | **0.0225 ±0.0021** | **0.0240 ±0.0039** | **0.0252 ±0.0072** |

(a) *Scaling experiment*

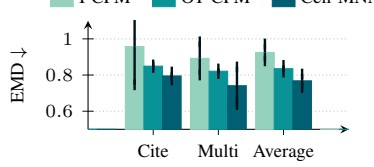

(b) *Amortization experiment*

Figure 3: (**a**) Model comparison across the synthetically inflated datasets. We report mean ± standard deviation of the MMD metric, along with the average across datasets. Lower values indicate better performance. Standard deviation is computed over left-out time points.(**b**) Comparison of models jointly trained on *Cite* and *Multi* datasets to test potential for amortization. We report mean ± standard deviation of the EMD metric, along with the average across datasets.

Foundation models have shown strong transfer learning capabilities across datasets in a variety of domains (Bodnar et al., 2025; Pearce et al., 2025; Bodnar et al., 2025). However, current SOTA methods for single-cell interpolation, such as OT-MFM and DeepRUOT, rely on multi-stage training or dataset-specific regularizers, making them less suitable for building foundation models. In contrast, the end-to-end nature of Cell-MNN enables amortized training across multiple datasets. We design an experiment to assess which models are promising for amortized training in the single-cell interpolation setting by jointly training on datasets with the same time scale, namely Cite and Multi.

**Training.** Our amortized training setup follows the single-cell interpolation experiment described in Section 4.1, with the only differences being that (i) we iteratively sample batches from Cite and Multi, (ii) we use a wider network with width 128, and (iii) we pass an additional dataset index into the model. We do not sample from the marginals at the left-out time point for either dataset. Since each dataset contains a different set of genes, we use the same PCA embeddings as in the previous experiment (Section 4.1) and merge datasets in the PCA subspace.

**Baselines.** We use OT-CFM as a baseline as it is the best-performing alternative model on the single-cell interpolation task that involves only a single training stage, making it easy to adapt to the amortized training setting. For each dataset, we compute the OT map on the entire dataset separately to ensure that the derived velocity labels are accurate. We use the hyperparameters specified by Tong et al. (2024a) and first reproduce the original results for the separate-dataset setting to verify our setup. In amortized training, we find that passing the dataset index as input does not affect OT-CFM's performance. For additional reference, we also report the performance of I-CFM.

**Results.** As shown in Figure 3b, Cell-MNN outperforms both OT-CFM and I-CFM in the amortized setting and achieves performance comparable to training on each dataset separately. Since the gene sets differ between datasets, transfer learning may be difficult in this setup. Nevertheless, these results suggest that for datasets with shared structure, Cell-MNN could enable transfer learning.

## 4.3 SCALABILITY AND ROBUSTNESS TO NOISE

Beyond leveraging multiple datasets to train a single model, the practical usefulness of a method depends on its ability to handle the increasingly large datasets available in the single-cell dynamics domain. In this context, performing OT preprocessing over all samples from two consecutive days, as required by OT-CFM, DeepRUOT, or OT-Interpolate, can become a significant bottleneck due to the quadratic time and space complexity of the Sinkhorn algorithm (Cuturi, 2013). To experimentally compare the scalability of different methods, we conduct the following scaling experiment.

**Training.** We synthetically inflate the dataset size of EB, Cite, and Multi to 250,000 cells each by resampling from each dataset and adding noise drawn from $\mathcal{N}(0, 0.1)$ to the PCA embeddings. Cell-MNN is trained with the same hyperparameters as in our first experiment in Section 4.1.

**Baselines.** We run OT-CFM and DeepRUOT on the inflated datasets and observe that both methods encounter *out-of-memory* (OOM) errors on our hardware (NVIDIA GeForce RTX 2080 Ti per model, 11 GB RAM) due to the quadratic memory complexity of the Sinkhorn algorithm. To mitigate this, Tong et al. (2024a) proposed a minibatch variant of optimal transport for OT-CFM, which achieves competitive image generation quality compared to dataset-wide OT preprocessing. We therefore use this mini-batch version as a baseline, denoted Batch-OT-CFM. As I-CFM does not require OT preprocessing, we also train it on the inflated datasets and report its performance.

**Metric.** For the larger datasets, computing the EMD metric becomes impractical, as it also requires estimating the OT map. We therefore use the MMD metric with a Laplacian kernel to compute the validation score. Since MMD can be computed in a batch-wise fashion, it is more practical for this experiment. We use the same hyperparameters for the MMD metric as for the training loss.

**Results.** We present the validation scores for models trained on inflated datasets in Table 3a. Performing dataset-wide OT preprocessing, as required by standard OT-CFM or DeepRUOT, leads to OOM errors due to the quadratic space complexity of the Sinkhorn algorithm. Batch-OT-CFM outperforms I-CFM, demonstrating the gains from minibatch OT. However, Cell-MNN achieves the best performance on all three inflated datasets, highlighting its scalability and robustness to noise.

## 4.4 DISCOVERING GENE INTERACTIONS

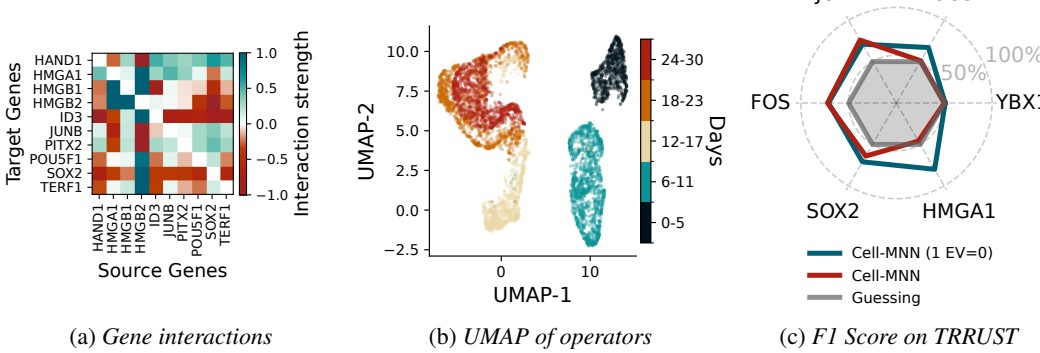

(a) *Gene interactions*    (b) *UMAP of operators*    (c) *F1 Score on TRRUST*

Figure 4: (**a**) Strongest predicted gene interactions by Cell-MNN for days 12–17 of the EB dataset, normalized to the range $[-1, 1]$. (**b**) UMAP projection of operators predicted by Cell-MNN on the EB dataset, showing that the model learns distinct dynamics at different time points. (**c**) Validation of predicted gene interactions by two Cell-MNN versions: For each source gene $j$, we classify each TRRUST edge $j \to i$ as activating or repressing using the sign of Cell-MNN's learned weight $w_{j \to i}$.

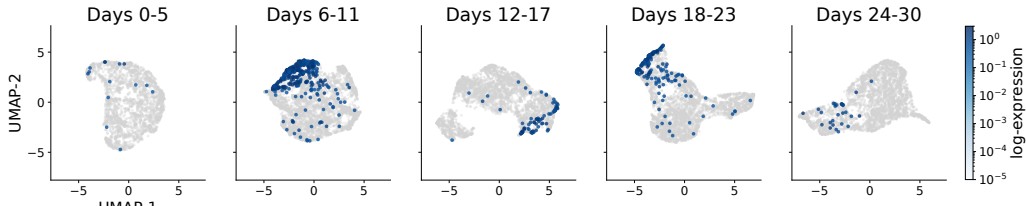

Figure 5: UMAPs of the predicted operators by Cell-MNN across the five time ranges of EB. Points are colored by whether joint expression of the EN-1 marker genes *FOXA2* and *SOX17* is above the 95th percentile. Clustering indicates that Cell-MNN learns distinct dynamics for the EN-1 cell type.

Cell-MNN predicts the local dynamics of the cell differentiation process in gene expression space, thereby learning interaction weights $w_{j \to i}$ from each gene to every other gene as described in Section 2.1. This corresponds to unsupervised learning of local gene interactions. To assess whether these learned interactions are biologically meaningful, we validate them against the literature-curated TRRUST database (Han et al., 2018), which contains 8,444 regulatory relationships synthesized from 11,237 PubMed articles. While TRRUST represents only a small subset of all potential relationships, it provides a valuable reference signal for evaluating Cell-MNN's predictions.

Table 2: GRN discovery on the EB dataset (F1 score in %). For each source gene we report the F1 score of predicting activation vs. repression for the $N_{int}$ labeled TRRUST interactions. Best method per row is highlighted in bold. Performance is averaged over three seeds for OT-CFM and Cell-MNN.

| Source gene | $N_{int}$ | SCODE | OT-CFM (J) | Cell-MNN (1EV=0) |
|---|---|---|---|---|
| JUN | 65 | 56.34% | 64% $\pm 2\%$ | **71%** $\pm 6\%$ |
| FOS | 25 | 38.10% | 50% $\pm 6\%$ | **71%** $\pm 10\%$ |
| YBX1 | 24 | 54.55% | **72%** $\pm 4\%$ | 51% $\pm 6\%$ |
| POU5F1 | 19 | 14.29% | 13% $\pm 1\%$ | **67%** $\pm 6\%$ |
| SOX2 | 16 | **84.21%** | 60% $\pm 0\%$ | 71% $\pm 9\%$ |
| HMGA1 | 10 | 25.00% | 31% $\pm 9\%$ | **80%** $\pm 6\%$ |
| Average | | 46% $\pm 25\%$ | 48% $\pm 22\%$ | **69%** $\pm 10\%$ |

**Unsupervised Classification.** Using labels from the TRRUST database, we design an *unsupervised* classification task for a source gene $j$: for every interaction $j \to i$ listed in TRRUST, we predict whether the relationship is activating or repressing. Since Cell-MNN outputs $w_{j \to i}$ per cell, we average these values over the dataset to obtain a single prediction for each interaction, classifying it as activating if $\sum_{(\boldsymbol{x},t) \in \mathcal{D}} w_{j \to i}(\boldsymbol{x}, t) > 0$ and repressing if smaller than zero. We report results for the genes $j$ that are in the top 10 most active ones (as by summing over all interaction weights

per gene) for any time point and that have more than 10 matching TRRUST interactions within the EB gene set. To evaluate Cell-MNN's capability of predicting the existence of interactions, we additionally performed an enrichment analysis which is presented in Appendix F.1.

**Training and Inductive Bias.** We train an ensemble of Cell-MNN models for single-cell interpolation on EB, each with a different left-out marginal. The setup matches Section 4.1 but uses a less preprocessed EB version, retaining gene names for downstream interaction analysis. To highlight the benefit of having access to an explicit dynamics model, we also introduce an additional model version with inductive bias on the parametrization of the linear operator $A_\theta$: in particular, we know that only some genes vary over time, implying that at least one eigenvalue of $A_\theta$ should be zero. During training, we therefore fix one eigenvalue to zero, forcing the model to learn static directions in the gene expression space. While this inductive bias reduces predictive performance in the single-cell interpolation setting by approximately $1\%$, it significantly improves gene interaction discovery performance on TRRUST (see Appendix 8 for ablation results).

**Baselines.** To contextualize the results, we focus our comparison to approaches that predict *signed* GRNs, as this is required for our classification task. As an ODE-based baseline, we run SCODE (Matsumoto et al., 2017) on the dataset. To compare with the idea of using the Jacobian of Neural ODEs as a proxy for the GRN (Qiu et al., 2022), we compute the Jacobians of a fully trained OT-CFM model (OT-CFM (J)). This can be thought of as an alternative to the operators predicted by Cell-MNN. Similar to Cell-MNN, we use the sign of the Jacobian to classify an interaction as activating or repressing. Given different models, we only adapt the GRN prediction in our pipeline; the rest of the evaluation remains the same.

**Results.** We compute precision, recall, and F1 scores for the *unsupervised* classification task, with all numerical values in Table 5 & 6. As shown in Table 2, Cell-MNN outperforms both SCODE and OT-CFM(J) by twenty percentage points in terms of F1 score, indicating that for the tested gene interactions, the model can meaningfully discover activation or repression in a fully unsupervised manner. Interestingly, the Cell-MNN variant with one eigenvalue set to zero improves classification performance from 60% to 69% in F1 score, demonstrating that the inductive bias introduced on the operator effectively constrains the solution space. Note that gene interactions are context-dependent (e.g., varying by cell type), and therefore the labels in TRRUST may not fully apply to the context of the EB dataset. Nevertheless, we view the agreement for the most dominant source genes as a meaningful signal that Cell-MNN has the potential to discover biologically plausible gene interactions.

**Visualizing Operators.** To visualize the learned operators, we plot UMAP projections of $A_\theta$ computed jointly across all time ranges (Figure 4b) and separately for each time range (Figure 5) in the EB dataset. The joint projection shows that Cell-MNN captures distinct dynamics across time, while the separate projections highlight differences between cell types. Additional UMAP visualizations for the cell types reported in the original EB study (Moon et al., 2019) are provided in Appendix H.

## 5 CONCLUSION

We introduced Cell-MNN, an encoder-decoder architecture whose representation is a locally linear latent ODE at the operating point of the cell differentiation dynamics. The model is able to perform trajectory inference, while *explaining* its predictions in terms of the underlying gene regulatory interactions used. Empirically, we show that Cell-MNN achieves competitive performance on single-cell benchmarks, as well as in scaling and amortization experiments. Importantly, the gene regulatory interactions learned from real single-cell data exhibit consistency with the literature-curated TRRUST database. Thus, Cell-MNN jointly addresses the challenges of trajectory reconstruction from snapshot data and gene interaction discovery.

Having shown that Cell-MNN learns biologically plausible gene interactions, a natural next step is to use it as a hypothesis generation engine for less-studied genes, guiding which interactions to test experimentally. Moreover, since Cell-MNN models dynamics in locally linear form, it may be possible to leverage the rich control theory literature on controller design for such locally linear systems. In principle, this could enable steering gene expression states toward desired configurations via perturbations, which could for example inform CRISPR-based gene edits (Jinek et al., 2012).

## ETHICS STATEMENT

All datasets used in this work (EB, Cite, Multi) are publicly available and were preprocessed by prior works Tong et al. (2024a; 2020). While the gene regulatory interactions predicted by Cell-MNN are partially validated against the TRRUST database, they should not be interpreted as definitive biological ground truth without further experimental validation. It is important to note that the model provides hypotheses for experimental follow-up, not direct medical recommendations. Insights from the model could eventually inform gene perturbation studies or therapeutic research. To mitigate potential misuse, we emphasize that the work is intended for advancing computational methodology in machine learning and computational biology, not for direct clinical application. We provide code and hyperparameters for reproducibility.

## REPRODUCIBILITY

We use the datasets in the form as preprocessed by prior work: Cite and Multi from Tong et al. (2024a)[2] and EB from Tong et al. (2020)[3]. For the gene discovery experiment, we use the less preprocessed version of EB provided by Tong et al. (2024a). All experiments are run on a single GPU and hyperparameters are presented in Table 3 in the Appendix together with the hardware and training time. We fix random seeds and report all results averaged over three runs with different seeds. The code is available on GitHub under `github.com/czi-ai/cell-mnn`, containing environment specifications in `environment.yml`. Commands to reproduce results are documented in the accompanying `README.md`.

## ACKNOWLEDGEMENTS

This work was supported by the Chan Zuckerberg Initiative (CZI) through the External Residency Program. We thank CZI for the opportunity to participate in this program, which enabled close collaboration and access to critical resources. We furthermore thank Peter Kharchenko, Bianca Dumitrascu, and Alicia Michael for very helpful discussions and guidance related to the single-cell application domain. We thank Alexander Tong and Lazar Atanackovic for sharing code and technical clarifications, which were essential for reproducing their results and conducting our subsequent experiments. Similarly, we thank the first authors of DeepRUOT (Zhang et al., 2025) and VGFM (Wang et al., 2025), Dongyi Wang and Zhenyi Zhang, for providing their code and assistance with its use. Finally, we thank the Causal Learning and Artificial Intelligence group at ISTA for their valuable feedback and discussions throughout the project.

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

## A  USE OF LARGE LANGUAGE MODELS (LLMS)

In this work, we used LLMs for (i) coding assistance during the software development phase, (ii) identifying relevant literature in response to specific research questions, and (iii) polishing and improving the readability of the paper. All substantive research contributions, analysis, and interpretations were carried out by the authors.

## B  DEFINITIONS

**Maximum Mean Discrepancy** (MMD, Gretton et al. (2012)): Given two distributions $p$ and $q$ over $\mathcal{X}$ and a positive-definite kernel function $k : \mathcal{X} \times \mathcal{X} \to \mathbb{R}$, the squared Maximum Mean Discrepancy (MMD) is defined as

$$\text{MMD}^2(p, q; k) = \mathbb{E}_{x,x' \sim p}[k(x, x')] + \mathbb{E}_{y,y' \sim q}[k(y, y')] - 2\,\mathbb{E}_{x \sim p, y \sim q}[k(x, y)].$$

## C  ARCHITECTURE

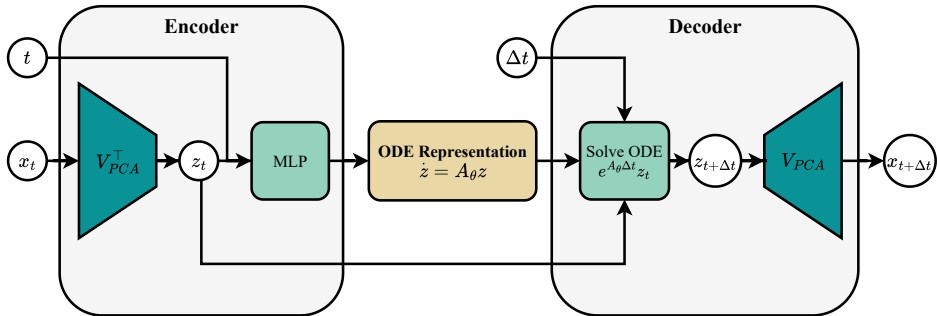

Figure 6: The Cell-MNN architecture first applies the PCA projection matrix to map the gene expression state $\boldsymbol{x}$ to a latent representation $\boldsymbol{z}_t$. An MLP then predicts a locally linear approximation $\dot{\boldsymbol{z}} = \boldsymbol{A}_\theta \boldsymbol{z}$ to the dynamics at the operating point $(\boldsymbol{z}_t, t)$. To decode, the analytical solution of this ODE is evaluated at a future time point and projected back into gene expression space.

## D  PROOFS

**Proposition 1** (Extension of Proposition 1 of Çimen (2010)). *Let $\boldsymbol{f} : \mathbb{R}^{d_z} \times \mathbb{R} \to \mathbb{R}^{d_z}$ satisfy $\boldsymbol{f}(\boldsymbol{0}, t) = \boldsymbol{0}$ for all $t \in \mathbb{R}$, and assume $\boldsymbol{f} \in \mathcal{C}^k(\mathbb{R}^{d_z} \times \mathbb{R})$ with $k \geq 1$. Then there exists a matrix-valued map $\boldsymbol{A} : \mathbb{R}^{d_z} \times \mathbb{R} \to \mathbb{R}^{d_z \times d_z}$ such that $\boldsymbol{f}(\boldsymbol{z}, t) = \boldsymbol{A}(\boldsymbol{z}, t)\,\boldsymbol{z}$ for all $(\boldsymbol{z}, t) \in \mathbb{R}^{d_z} \times \mathbb{R}$.*

*Proof.* Fix $(\boldsymbol{z}, t) \in \mathbb{R}^{d_z} \times \mathbb{R}$ and define $\gamma : [0, 1] \to \mathbb{R}^{d_z}$ by $\gamma(s) := \boldsymbol{f}(s \cdot \boldsymbol{z}, t)$. Since $\boldsymbol{f} \in \mathcal{C}^k$ and $k \geq 1$, the map $s \mapsto \gamma(s)$ is differentiable and

$$\frac{d}{ds}\gamma(s) = D_{\boldsymbol{z}}\boldsymbol{f}(s \cdot \boldsymbol{z}, t)\,\boldsymbol{z}.$$

By the fundamental theorem of calculus,

$$\boldsymbol{f}(\boldsymbol{z}, t) - \boldsymbol{f}(\boldsymbol{0}, t) = \gamma(1) - \gamma(0) = \int_0^1 D_{\boldsymbol{z}}\boldsymbol{f}(s \cdot \boldsymbol{z}, t)\,\boldsymbol{z}\,ds = \left(\int_0^1 D_{\boldsymbol{z}}\boldsymbol{f}(s \cdot \boldsymbol{z}, t)\,ds\right)\boldsymbol{z}.$$

Using $\boldsymbol{f}(\boldsymbol{0}, t) = \boldsymbol{0}$ gives $\boldsymbol{f}(\boldsymbol{z}, t) = \boldsymbol{A}(\boldsymbol{z}, t)\,\boldsymbol{z}$. $\qquad\qquad\square$

Table 3: Collection of training hyperparameters used in all experiments. All models were trained on a single NVIDIA RTX 2080 Ti GPU. Training on EB, Cite, and Multi separately, as well as on the amortization experiment, required about 1 hour per run, while training on inflated datasets took roughly 4 hours. No distributed training or large-scale compute resources were required.

| Component | Hyperparameter |
|---|---|
| Data preprocessing | PCA projection |
| MLP ($\boldsymbol{A}_\theta$) | Depth: 4 |
| | Width: 96 (128 for amortization experiment) |
| | Activation: Leaky ReLU |
| | Initialization: Kaiming normal |
| | Last layer scale: 0.01 |
| MMD kernel | Laplacian kernel $k(z, z') = \exp[-\frac{\max(||z-z'||_1, \epsilon)}{\sigma d_z}]$ |
| | $\sigma = 1, \epsilon = 10^{-8}$ |
| Optimization | Batch size per time point: 200 |
| | Future discount factor $\gamma = 0.1$ |
| | Initialization scale: 0.01 |
| | Regularization $\lambda_{\mathrm{kin}} = 0.1, \lambda_{\mathrm{inv}} = 1$ |
| | Optimizer: AdamW |
| | Learning rate: $2 \times 10^{-4}$ |
| | Weight decay: $1 \times 10^{-5}$ |
| Validation | Frequency: every 10 steps |
| | Patience: 40 validation checks |
| Training time | 60 minutes (240 minutes for inflated datasets) |
| Randomness | Seeds: 3 |
| Hardware | $1\times$ NVIDIA GeForce RTX 2080 Ti (11 GB RAM) |

# E ON THE USE OF LOW DIMENSIONAL PCA EMBEDDINGS

**Low-dimensional PCA as standard practice.** In line with prior work on single-cell trajectory inference, we operate in a low-dimensional latent space obtained by PCA on the gene expression matrix. Concretely, we project each dataset to the first five principal components and train both Cell-MNN and all baselines in this common representation. This choice follows the prevailing assumption in computational biology that scRNA-seq data lie near a low-dimensional manifold, and it is consistent with the implementation of *nine* previous works reported in Table 1.

**Empirical validation of the 5D PCA representation.** Figure 7 provides a quantitative and qualitative assessment of the 5D PCA space used in our experiments. Figures 7a, 7e and 7e show that the first few principal components account for the large part of the variance in each dataset: using five components yields a cumulative explained variance above 60% for Cite and Multi and above 40% for Embryoid. Importantly, this low-dimensional representation preserves cell-type structure. We compute $k$-nearest neighbor classification in the 5D PCA space with $k = 15$ and obtain accuracies of 87% (Multi) and 90% (Cite); for Embryoid, cell-type labels are not available. Consistently, the UMAPs computed from the 5D PCA embedding (Figures 7b and 7d) exhibit clear clustering by cell type, indicating that the embedding retains the information required to distinguish cellular lineages. To show flexibility with respect to the number of principle components used, we also train Cell-MNN and OT-CFM in 10 dimensional PCA subspace with no tuning and find that they perform similarly, see Table 10.

**Scope of our contribution.** Our focus in this work is orthogonal to learning the optimal representation: We focus on improving the dynamical model given a standard low-dimensional embedding. Within the 5D PCA space, Cell-MNN achieves SOTA average performance on single-cell interpolation benchmarks (Table 1), while removing OT preprocessing and learning explicit, local ODEs that can be interpreted as gene interactions. For scientific questions centered on lineage bifurcations and fate decisions, the key requirement is that the representation preserves cell-type structure. The experiments in Figure 7 show that this requirement is met. Exploring richer or alternative represen-

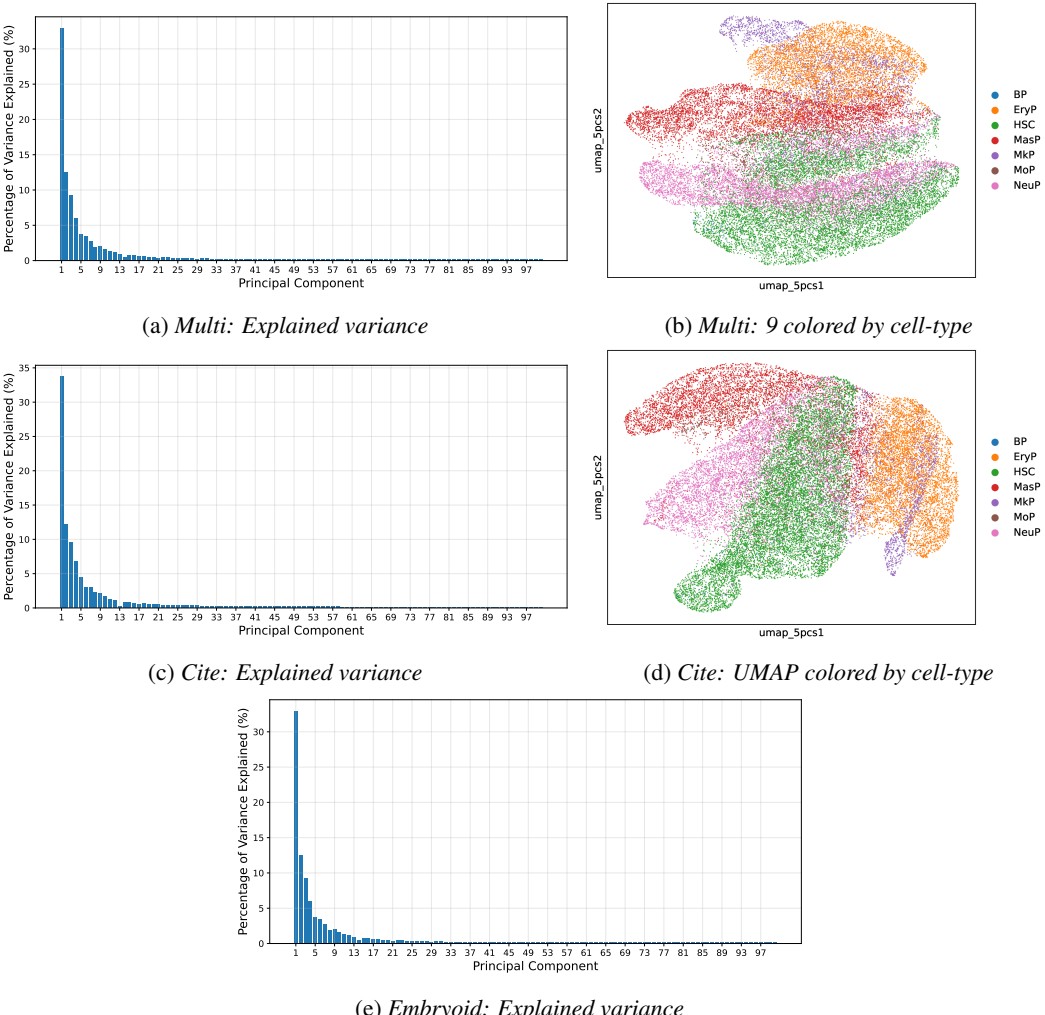

(a) *Multi: Explained variance*

(b) *Multi: 9 colored by cell-type*

(c) *Cite: Explained variance*

(d) *Cite: UMAP colored by cell-type*

(e) *Embryoid: Explained variance*

Figure 7: (**a, c, e**) Explained variance of each of the principle components plotted for the datasets Multi, Cite and Embryoid. The first few components capture the majority of the variance. (**b, d**) UMAPs computed on the 5 dimensional PCA embedding colored by cell-type. Clustering by cell-type shows that 5 dimensional PCA embedding retains cell-type information.

tations is an interesting orthogonal direction, and Cell-MNN can in principle be applied on top of such alternatives without changing the core methodology.

Table 4: Gene selection specifications for the TRRUST experiment. There are 16 genes that make up the top 10 predicted high-interaction source genes across the five time points, Of these, 14 are contained in TRRUST, and 6 have more than 10 interactions overlapping with the training gene set. This table provides shows how source genes were selected for downstream evaluation.

| Top Source Gene | In TRRUST | # Interactions in TRRUST | # in Training Gene Set | $> 10$ Interactions |
|---|---|---|---|---|
| HMGA1 | True | 18 | **10** | True |
| HMGB2 | True | 2 | | |
| JUNB | True | 15 | 4 | |
| FOS | True | 63 | **25** | True |
| JUN | True | 173 | **65** | True |
| POU5F1 | True | 25 | **19** | True |
| HAND1 | False | 0 | 0 | |
| ID2 | True | 2 | | |
| TERF1 | True | 1 | | |
| PITX2 | True | 11 | 4 | |
| ID3 | True | 2 | | |
| HMGB1 | False | 0 | 0 | |
| SOX2 | True | 23 | **16** | True |
| HMGA2 | True | 5 | | |
| YBX1 | True | 33 | **24** | True |
| ID1 | True | 1 | | |

Table 5: Validation of predicted gene interactions on TRRUST: For each source gene $j$, we classify each TRRUST edge $j \rightarrow i$ as activating or repressing using the sign of the learned weight $w_{j \rightarrow i}$ from Cell-MNN with one eigenvalue set to zero (averaged over cells). For each source gene, we report the number of interactions in TRRUST and classification metrics (precision, recall, and F1) shown as mean $\pm$ std across ensemble models trained on three different seeds.

| Source Gene | # Interactions $\downarrow$ | Precision | Recall | F1 |
|---|---|---|---|---|
| JUN | 65 | $62\% \pm 8\%$ | $82\% \pm 3\%$ | $71\% \pm 6\%$ |
| FOS | 25 | $65\% \pm 10\%$ | $80\% \pm 10\%$ | $71\% \pm 10\%$ |
| YBX1 | 24 | $55\% \pm 10\%$ | $48\% \pm 3\%$ | $51\% \pm 6\%$ |
| POU5F1 | 19 | $82\% \pm 6\%$ | $58\% \pm 11\%$ | $67\% \pm 6\%$ |
| SOX2 | 16 | $73\% \pm 12\%$ | $69\% \pm 8\%$ | $71\% \pm 9\%$ |
| HMGA1 | 10 | $78\% \pm 9\%$ | $82\% \pm 2\%$ | $80\% \pm 6\%$ |

# F  GENE REGULATORY INTERACTION RECOVERY

To quantitatively assess the learned gene interactions, we designed an unsupervised classification task based on the TRRUST database, which contains literature-curated gene regulatory interactions, many of which are annotated as activating or repressing. For evaluation of our model, we focus on the most dominant *source genes* predicted by Cell-MNN, i.e., those with the highest mean interaction strength with other genes. A source gene is included into the experiment if at least 10 of its interactions are listed in TRRUST. For each such gene, we classify the direction of its effect on downstream targets as activating or repressing. Since Cell-MNN produces cell-specific predictions of interaction weights, we average these over 10,000 cells to obtain a robust prediction for each interaction. Based on these predictions, we compute precision, recall, and F1 scores to quantify how well the model recovers known regulatory mechanisms and report them in Table 5 and Table 6.

Table 6: Ablation of models with *no* eigenvalue forced to zero on predicting gene interactions on TRRUST as in Table 5.

| Source Gene | # Interactions ↓ | Precision | Recall | F1 |
|---|---|---|---|---|
| JUN | 65 | $69\% \pm 11\%$ | $86\% \pm 7\%$ | $76\% \pm 10\%$ |
| FOS | 25 | $66\% \pm 22\%$ | $79\% \pm 19\%$ | $72\% \pm 21\%$ |
| YBX1 | 24 | $58\% \pm 11\%$ | $45\% \pm 8\%$ | $50\% \pm 8\%$ |
| POU5F1 | 19 | $56\% \pm 39\%$ | $48\% \pm 26\%$ | $51\% \pm 32\%$ |
| SOX2 | 16 | $67\% \pm 12\%$ | $62\% \pm 4\%$ | $64\% \pm 8\%$ |
| HMGA1 | 10 | $47\% \pm 31\%$ | $47\% \pm 34\%$ | $46\% \pm 33\%$ |

Table 7: Amortized model comparison across the *Cite* and *Multi* datasets. We report mean $\pm$ standard deviation of the EMD metric, along with the average across datasets. Lower values indicate better performance. Standard deviation is computed over left-out time points.

| Model | Cite | Multi | Average ↓ |
|---|---|---|---|
| I-CFM (Tong et al., 2024a) | $0.957 \pm 0.211$ | $0.892 \pm 0.092$ | $0.925 \pm 0.047$ |
| OT-CFM (Tong et al., 2024a) | $0.849 \pm 0.007$ | $0.821 \pm 0.013$ | $0.835 \pm 0.019$ |
| Cell-MNN | $\mathbf{0.795 \pm 0.022}$ | $\mathbf{0.741 \pm 0.104}$ | $\mathbf{0.768 \pm 0.038}$ |

Table 8: Cell-MNN ablation study on single-cell interpolation benchmark when setting one eigenvalue (EV) of $\boldsymbol{A}_\theta$ to zero. Average predictive performance degrades by less than $1\%$.

| Method | Cite | EB | Multi | Average ↓ |
|---|---|---|---|---|
| Cell-MNN (One EV$= 0$) | $0.795 \ _{\pm 0.016}$ | $0.701 \ _{\pm 0.076}$ | $0.746 \ _{\pm 0.097}$ | $0.748 \ _{\pm 0.049}$ |
| Cell-MNN (All EVs predicted) | $\mathbf{0.791} \ _{\pm \mathbf{0.022}}$ | $\mathbf{0.690} \ _{\pm \mathbf{0.073}}$ | $\mathbf{0.742} \ _{\pm \mathbf{0.100}}$ | $\mathbf{0.741} \ _{\pm \mathbf{0.050}}$ |

## F.1 PREDICTING THE EXISTENCE OF INTERACTIONS

As an additional validation of the gene interactions predicted by Cell-MNN, we evaluate its performance at predicting the existence of regulatory links. To this end, we first rank all predicted interactions by their inferred strength and then assess this ranking against the TRRUST database. To restrict TRRUST to interactions that are plausibly involved in the differentiation dynamics of the EB dataset, we subset the database to interactions whose transcription factor (TF) regulator is mentioned as relevant in the original analysis of Moon et al. (2019) (Fig. 6d). This yields 447 interactions regulated by 70 TFs, which we treat as our ground-truth signal. This restriction is also

Table 9: Comparison of GRN discovery methods on the EB dataset. We report precision@500 and AUROC for predicting existence of TRRUST interactions. Higher is better. All metrics were computed over three seeds.

| Method | Enrichment@500 | AUROC |
|---|---|---|
| GRNBoost2 | 20.429 | 0.633 |
| SCODE | 16.714 | 0.686 |
| OT-CFM (J) | 20.429 | 0,661 |
| Cell-MNN (ours) | 18.572 | 0.659 |

necessary to keep the evaluation of the baselines computationally tractable. We define the candidate interaction set as all directed TF-target pairs where the TF is among the 70 EB regulators and the target is any gene in the EB dataset.

**Baselines.** As this experiment only requires a ranking of interactions, we can compare against methods that predict *unsigned* GRNs. We therefore include the widely used GRN discovery methods GRNBoost2 and SCODE as baselines. We also compute the performance of OT-CFM (J) on this task. Because we are only interested in the presence of a regulatory link, we evaluate all methods on the absolute interaction strength, ignoring the sign of the effect.

**Evaluation and Metrics.** For each method, we obtain a scalar interaction score for every TF-target pair in the candidate set, and assemble these into a TF-target score matrix in the common gene space. For Cell-MNN, interaction scores are derived from an average over 100 operators from

each time point in the dataset. We use an ensemble of three Cell-MNN models each trained with a different left-out time point. We evaluate all methods using AUROC and Precision@K. The AUROC measures how well a method separates interacting from non-interacting pairs across all possible thresholds, whereas Precision@K measures the proportion of true interactions among the top-$K$ ranked edges. To make Precision@K more interpretable, we normalize it by the base precision of a uniform ranking over all candidate interactions and refer to the resulting quantity as Enrichment@K. Intuitively, Enrichment@K measures by which factor a method outperforms random guessing.

**Results and Discussion.** The results of this experiment are summarized in Table 9. We find that Cell-MNN performs competitively with the baselines in terms of both AUROC and Enrichment@K. We note that we restrict GRNBoost2 to learning interactions only for the 70 EB regulators, which effectively provides them with additional prior knowledge. For both Cell-MNN and OT-CFM, we also substantially coarse-grain their outputs by averaging interaction scores across cells and discarding sign information. Consequently, both methods, which in principle can produce context-dependent, signed predictions, are evaluated here in a much more restricted, global setting.

## G    ADDITIONAL INTERPOLATION RESULTS

Table 10: Model comparison on a 10-dimensional PCA embedding. We report the mean $\pm$ standard deviation of the EMD metric across the *Cite*, *EB*, and *Multi* datasets, along with the average across datasets. Lower values indicate better performance.

| Method | Cite (10D) | EB (10D) | Multi (10D) | Average (10D) |
|---|---|---|---|---|
| OT-CFM | $1.491 \pm 0.013$ | $1.607 \pm 0.074$ | $1.678 \pm 0.248$ | $1.592 \pm 0.112$ |
| Cell-MNN (ours) | $1.502 \pm 0.012$ | $1.587 \pm 0.113$ | $1.709 \pm 0.177$ | $1.599 \pm 0.101$ |

We provide further numerical results complementing the main experiments. For the single-cell interpolation task (Section 4.1), Table 8 reports an ablation in which the model is trained with one eigenvalue set to zero, as later used in the gene interaction discovery experiment. Table 7 presents the results of the amortization experiment across datasets (Section 4.2).

### G.1    HIGH DIMENSIONAL EXPERIMENTS

In this section, we evaluate the performance of Cell-MNN in higher-dimensional latent spaces. To this end, we train the same model, with slightly modified hyperparameters, in 50- and 100-dimensional PCA subspaces. Following Neklyudov et al. (2024), we do not whiten the data in PCA space to preserve the empirical variance structure. To keep feature magnitudes in a numerically well-conditioned range for MLP training while preserving their relative variance, we rescale all components by the standard deviation of the first principal component. We find that this improves the stability of training Cell-MNN. Since the EMD is homogeneous under a global rescaling of both distributions ($\mathrm{EMD}(\lambda \boldsymbol{X}, \lambda \boldsymbol{Y}) = \lambda \, \mathrm{EMD}(\boldsymbol{X}, \boldsymbol{Y}), \, \lambda > 0$), we multiply the EMD scores computed on the rescaled PCA coordinates by the standard deviation of the first principal component so that they are comparable to those obtained on the original (unscaled) PCA space.

Due to increased RAM requirements, we use a different GPU, namely an NVIDIA L40S (48 GB RAM). This also allows us to train with a larger batch size of $1028$. We set the learning rate to $1 \times 10^{-3}$ (50D) and $5 \times 10^{-5}$ (100D), patience for early stopping to 10 evaluation steps and keep the remaining hyperparameters unchanged. We train on the Cite and Multi and find that the runtimes range from 4m 25s to 31m 29s, depending on the seed and left-out time point. Memory usage remains below 25 GB of RAM in this setup.

We report the results of the experiments in Table 11. Without tuning the hyperparameters further, we find that Cell-MNN performs within error bars of SOTA approaches for the Multi dataset.

We remark that, due to the analytical solution of the ODE, one can choose to decode the trajectories at fewer time discretization points without impacting the accuracy of the predicted trajectories. This can be used to reduce the RAM requirements of the method and is unique when compared to Neural ODEs, whose accuracy depends on the step size due to numerical solving.

Table 11: Single-cell interpolation on 50- and 100-dimensional PCA embeddings across the *Cite* and *Multi* datasets. We report the mean $\pm$ standard deviation of the EMD metric computed over three seeds. Values marked * are computed by us.

| Method | Cite (50D) | Multi (50D) | Cite (100D) | Multi (100D) |
|---|---|---|---|---|
| I-CFM | 41.834 $\pm$3.284 | 49.779 $\pm$4.430 | 48.276 $\pm$3.281 | 57.262 $\pm$3.855 |
| WLF-SB | 39.695 $\pm$1.935 | 47.828 $\pm$6.382 | 46.131 $\pm$0.083 | 55.065 $\pm$5.499 |
| WLF-OT | 38.352 $\pm$0.203 | 47.890 $\pm$6.492 | 44.821 $\pm$0.126 | 55.416 $\pm$6.097 |
| OT-CFM | 38.756 $\pm$0.398 | 47.576 $\pm$6.622 | 45.393 $\pm$0.416 | 54.814 $\pm$5.860 |
| $[SF]^2$M-Exact | 40.009 $\pm$0.783 | 45.337 $\pm$2.833 | 46.530 $\pm$0.426 | 52.888 $\pm$1.986 |
| $[SF]^2$M-Geo | 38.524 $\pm$0.293 | 44.795 $\pm$1.911 | 44.498 $\pm$0.416 | 52.203 $\pm$1.957 |
| WLF-UOT | 37.007 $\pm$1.200 | 46.286 $\pm$5.841 | 43.731 $\pm$1.375 | 54.222 $\pm$5.827 |
| Cell-MNN (ours)* | 38.803 $\pm$0.635 | 43.926 $\pm$2.590 | 46.020 $\pm$1.177 | 52.698 $\pm$2.341 |
| OT-CFM* | 38.576 $\pm$0.429 | 43.141 $\pm$3.918 | 45.368 $\pm$0.473 | 51.399 $\pm$3.972 |
| OT-MFM | 36.394 $\pm$1.886 | 45.160 $\pm$4.960 | 41.784 $\pm$1.020 | 50.906 $\pm$4.627 |

## H  ADDITIONAL QUALITATIVE RESULTS

In Figures 8, 9, 10, 11, we present UMAP projections of the learned operators for each time range, colored by all the cell types reported in the developmental graph of Moon et al. (2019). These correspond to the same UMAPs described in Section 2.1, recolored by different cell type to highlight the cell-type dependence of the predicted dynamics. Cells are assigned to a type when the joint expression of the associated marker genes exceeds the 95th percentile. This analysis is enabled by having access to an explicit dynamics model conditioned on time and gene expression, which potentially allows inferences such as identifying when two cell types share similar dynamical laws within a given time range.

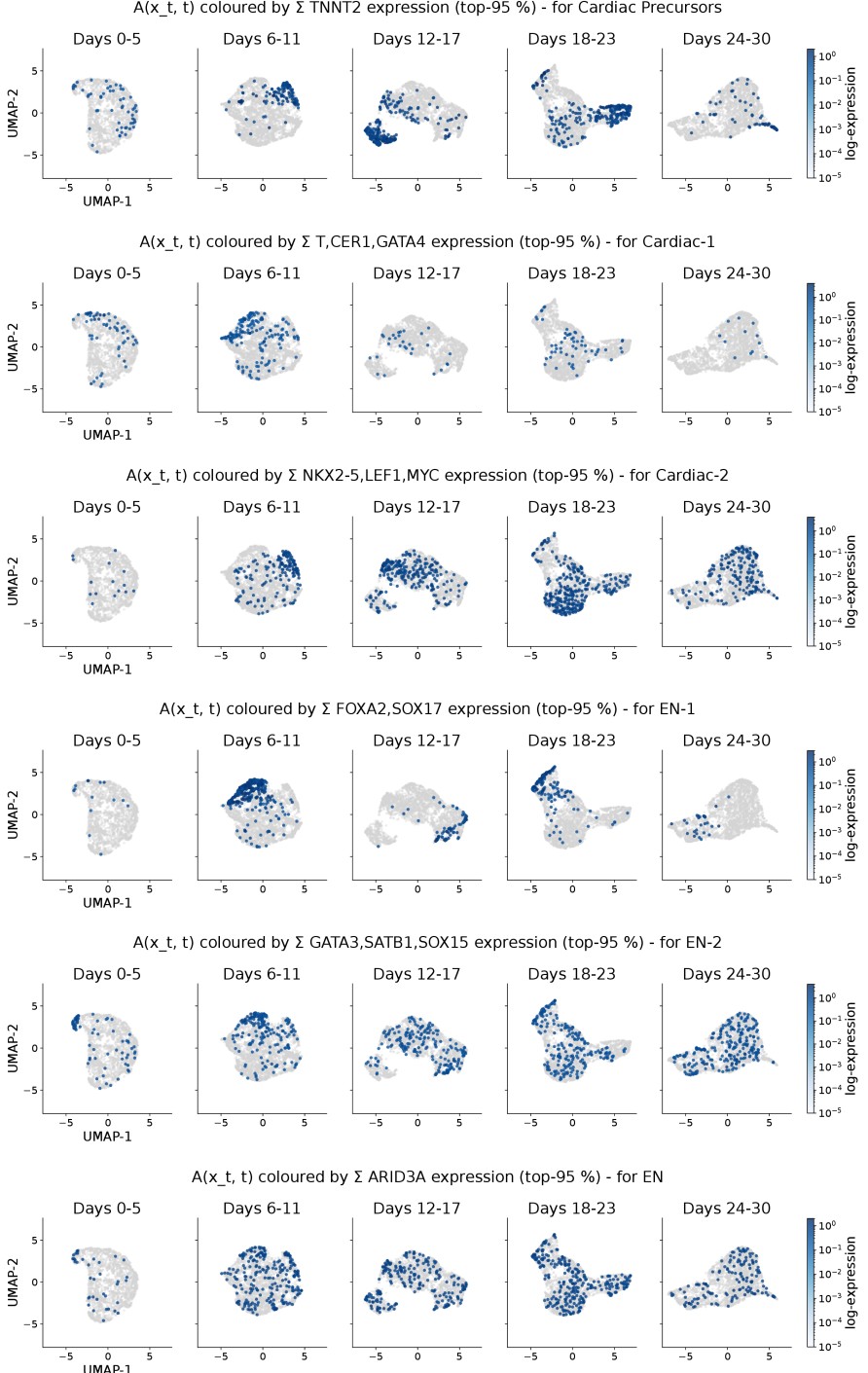

Figure 8

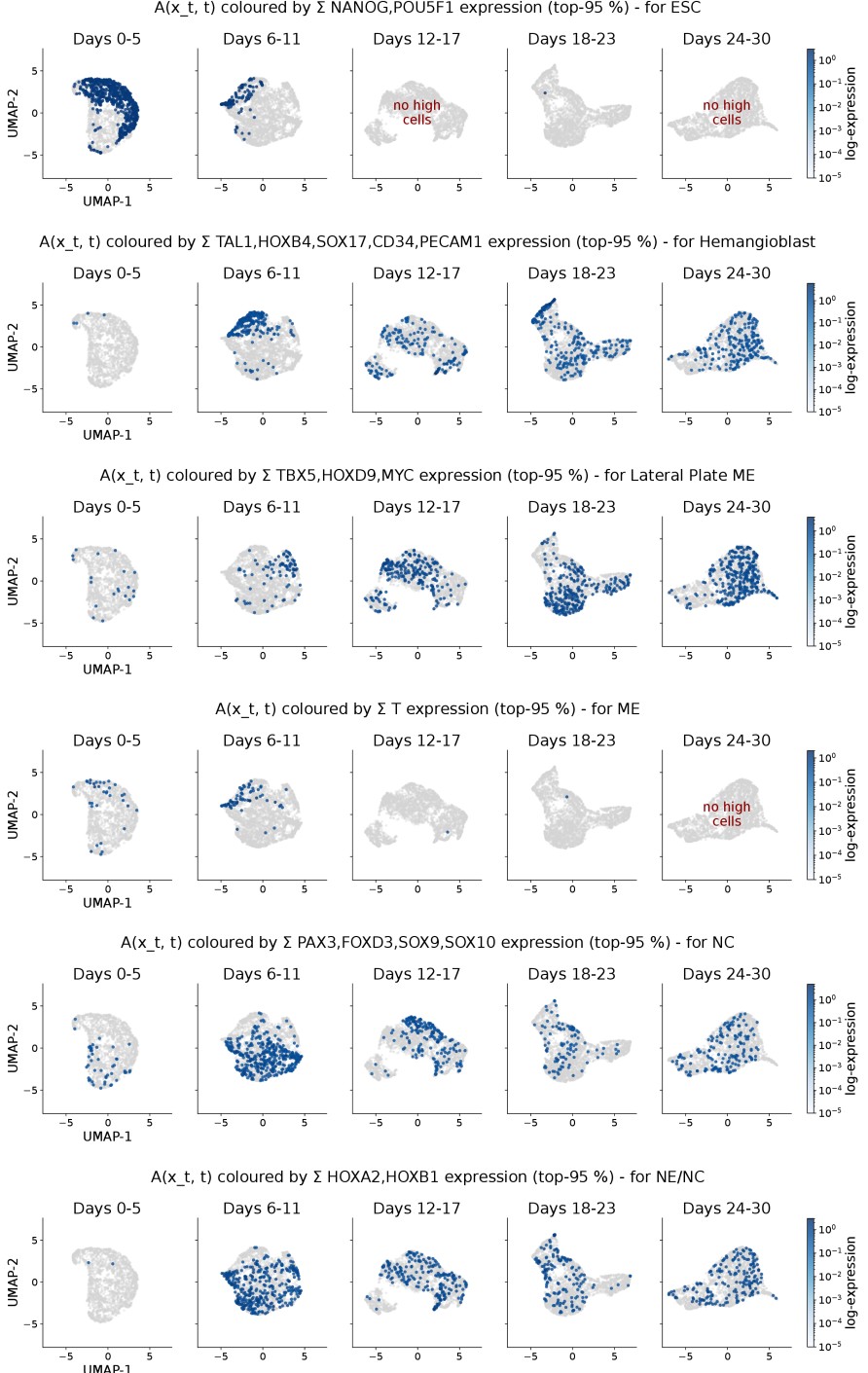

Figure 9

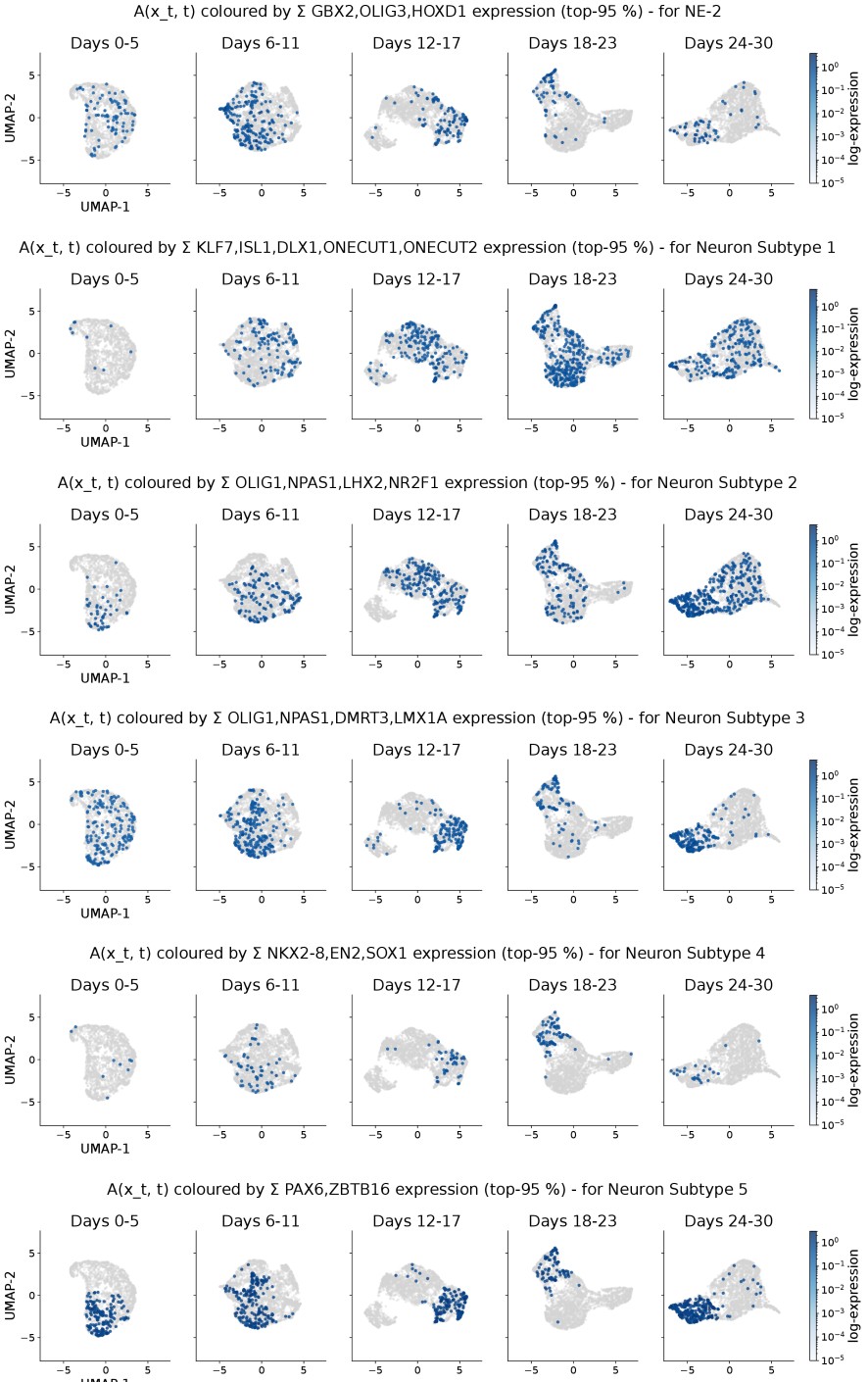

Figure 10

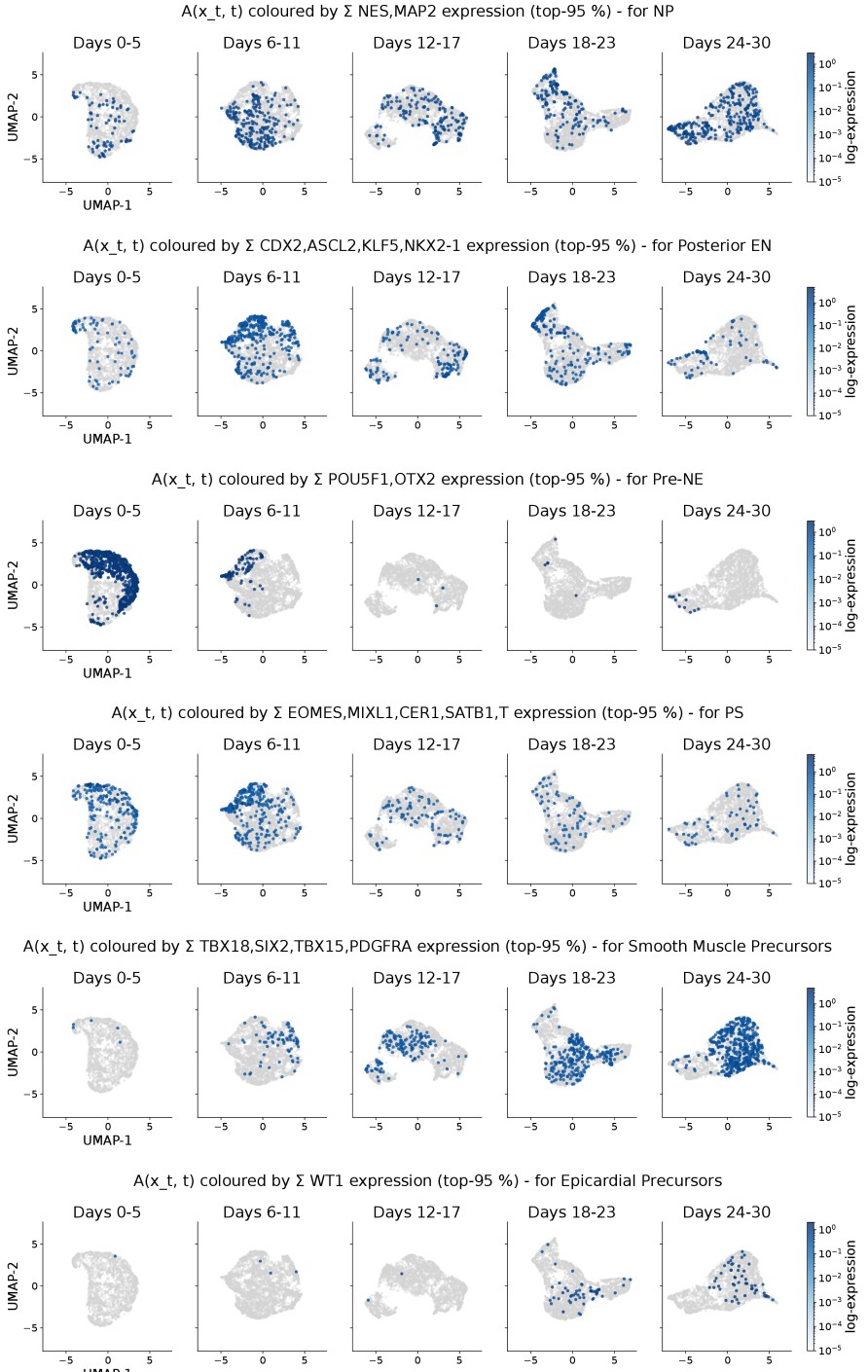

Figure 11

