# OpenReview forum: "Learning Explicit Single-Cell Dynamics Using ODE Representations"
_ICLR.cc/2026/Conference — ICLR 2026 Poster_

### Official Review · Reviewer_JgxX · 2025-10-24

**Soundness:** 3
**Presentation:** 3
**Contribution:** 3
**Rating:** 6
**Confidence:** 3

**Summary:**

The authors propose to use a variant of Neural ODEs where the velocity is parameterized as a constant linear function of location at every timestep.  They use this parameterization to fit single-cell RNA data over several timepoints, evaluating with interpolation of a heldout timepoint.

**Strengths:**

The authors show good results compared to other interpolation methods on several common benchmarks.  The motivation for the method is straightforward and natural, and I appreciate the work in making simpler and more interpretable models competitive with flow-based trajectory learning.

**Weaknesses:**

Projecting to 5d PCA is a considerable limitation, as the method seems to not scale and the dynamics of single cell data is generally not so easily compressible.  Additionally, the actual training of the model is somewhat unclear (see questions).

I’m also not totally convinced by the gene analysis results, as it probably requires a baseline to convince a reader that these top gene interactions were not simply the ones encoded by PCA, and any model that learns a velocity (say, MFM-OT or TrajectoryNet) could linearize its velocity at some timepoint and find the same gene interactions.  The later claim about enforcing a zero in the spectrum seems a bit flimsy; it’s true that many individual genes will not be changing during the dynamics, but at only 5 PCs one would expect all PCs were relevant to the dynamics.

**Questions:**

How exactly is the model trained?  The locally linear ODE maps from z_t to z_{t + \Delta_t} but I cannot find an indication of how the authors turn this into an explicit loss.  They make no mention of differentiating through an ODE solver so is there a parameter \Delta_t set in the method and the authors explicitly differentiate through several linearization steps?

---

> ### Author Response · Authors · 2025-11-20
> **Answer Reviewer JgxX**
>
> We thank the reviewer for their feedback and are very happy to hear that they find *“... the method is straightforward and natural”* and that they *“...appreciate the work in making simpler and more interpretable models competitive with flow-based trajectory learning”*.
>
> # Answer to concerns
> The reviewer has two major concerns, which were actually shared among most of the reviewers: (i) the use of only five principle components and (ii) the additional baselines in the GRN discovery experiment. As they are shared among reviewers, we address them in the general statement. On (i), we show that Cell-MNN outperforms two suitable baselines (SCODE and Jacobian of OT-CFM). As a response to (ii), we perform canonical analysis of the data to show that the first 5 principle components capture the most essential information for scientific application (cell-type). We kindly redirect the reviewer to this general response for detailed answers to their two concerns.
>
> # Questions on *“How exactly is the model trained?”*
>
> *“…cannot find an indication of how the authors turn this into an explicit loss”*
>
> The details on the loss are described in the Optimization paragraph in *Section 2.1*: We indeed map $z_t$ to $z_{t + \Delta t}$ using the locally linear ODE. We train the model on a combination of MMD loss on the marginal distributions and kinetic energy regularizer on the trajectories. For quick intuition: We predict $z_{t + \Delta t}$ for all input samples of a batch and then use a batch of samples from the empirical marginal distribution at $t + \Delta t$ as ground truth signal to compute the MMD on.
>
> *“... so is there a parameter $\Delta t$ set in the method and the authors explicitly differentiate through several linearization steps?”*
>
> As there exists an analytical solution to the locally linear ODE, we do not need to employ a numerical solver. The analytical solution can just be evaluated and it is furthermore differentiable, which makes it straightforward to implement using autodifferentiation. This is specifically possible because we only predict the operator once per input sample and then keep it constant to predict the marginal distribution at the next time point. The $\Delta t$ is essentially defined by the time difference by the time labels in the datasets $\Delta t_{1} = t_2 - t_1$.
>
> # Closing remarks
> We hope that these explanations make the model training more intuitive to the reviewer. If a particular point has particularly helped the reviewer's intuition, we would highly appreciate it if they could circle it back such that we can improve our explanations in the paper.

---

> > ### Comment · Reviewer_JgxX · 2025-11-25
> >
> > Thank you for your response.  On the issue of 5 PCs being enough, I'm not entirely convinced; for lineage inference of harder tasks, for example time-series of zebrafish embryos as in [1], I'm skeptical that 5 PCs would capture many lineages being inferred simultaneously.  Likewise, for GRN inference I would've liked to have seen the comparison to methods like GRNBoost or Genie3 on edge existence rather than edge direction, as these methods are the standard.
> >
> > I appreciate the authors clearer description of their method.  In that case for clarity it might help to emphasize in the notation that \Delta t always corresponds to t_{i+1} - t_i in practice.  I will keep my positive score.
> >
> > [1] Saunders, Lauren M., et al. "Embryo-scale reverse genetics at single-cell resolution." Nature 623.7988 (2023): 782-791.

---

> > > ### Author Response · Authors · 2025-11-26
> > >
> > > We thank the reviewer for their reply and additional feedback.
> > >
> > > **Higher dimensional latent spaces.** We agree with the reviewer that the number of required PCs is task dependent. In response to this point, we have worked on evaluating the method's extendibility to 50 dimensional and 100 dimensional PCA embeddings and find that Cell-MNN performs competitively also in these contexts. We give a quick summary of the results in a new general message. Please refer to **Appendix F.1** of the paper for all the details.
> > >
> > > **GRN edge existence prediction.** We also find performance at predicting edge existence to be a desirable property to measure. However, we have to note that the TRRUST database only contains information about the *existence* of links and not their *absence*, which is why we find it non-trivial to design a rigorous experiment for edge existence. In the following we will suggest two alternative experiments that we think could work in this setting:
> > >
> > > 1. **Unsigned GRN method in existing experiment.** We could run a method such as GRNBoost on the data to predict an unsigned GRN. To then compare with our method, we would label the unsigned GRN by the signs contained in the TRRUST database and evaluate how accurate the baseline is at predicting the known interactions. This way we could include unsigned GRN discovery approaches in our existing experiment.
> > > 2. **Enrichment analysis of top interactions.** An alternative to the one above would be to first select a set of genes that are known to influence the differentiation process given the original paper (Moon et al. 2019). We could then rank the predicted interactions of these genes, and, for some threshold, measure the precision at predicting the interactions contained in TRRUST. We remark that this evaluation treats all absent interactions as negatives and therefore underestimates the true precision
> > >
> > > We would kindly ask the reviewer to respond whether either of the two experiments would address their concern or how to alter it otherwise.
> > >
> > > Thank you for your point on notation. We will add a short note on the use of $\Delta t$ in the paper to clarify this.

---

> > > > ### Comment · Reviewer_JgxX · 2025-11-27
> > > >
> > > > I agree that assessing GRN recovery is difficult, I would imagine experiments similar to those in [1] would be more convincing, with simulated GRNs from BoolODE and the Renge dataset.  Of course that is a concurrent work so I don't mean comparing directly to their method, but using their GRN recovery assessments as they are able to benchmark against methods like OTVelo, and it would be straightforward to try GRNBoost or other methods that don't require timepoints.
> > > >
> > > > [1] Rimawi-Fine, Noah El, et al. "Simulation-free Structure Learning for Stochastic Dynamics." arXiv preprint arXiv:2510.16656 (2025).

---

> > > > > ### Author Response · Authors · 2025-12-01
> > > > >
> > > > > We thank the reviewer for the suggestion and for understanding that this evaluation is difficult. After careful consideration, and to address the concerns of you and otWS jointly, we decided to perform an enrichment analysis of predicted links. We opt for an experiment on real data and use models trained on the Embryoid dataset and evaluate them with respect to known links in the TRRUST database. We use AUROC score and Precision@K to measure performance and find that Cell-MNN performs competitively compared to all baselines (GRNBoost2, SCODE and OT-CFM). Please refer to our new general message in which we give an experiment summary. For all the details, we point the reviewer towards **Appendix E.1**.

---

### Official Review · Reviewer_otWS · 2025-10-25

**Soundness:** 3
**Presentation:** 3
**Contribution:** 3
**Rating:** 6
**Confidence:** 4

**Summary:**

In this paper, the authors propose Cell-MNN, a new ODE-based method designed to model the single-cell dynamics of cellular differentiation. The method operates in PCA space, and uses a hypernetwork to learn the transition matrix A. The local dynamics are assumed to be linear. The model is trained by optimizing an MMD loss between the predicted and target marginals of gene expression in the latent space. They benchmark their model on single-cell interpolation, amortized training, and gene interaction directionality.

**Strengths:**

- The method eliminates the need for OT which is typically a computational bottleneck for many methods.
- The method achieves competitive performance on the cell interpolation task, outperforming multiple baselines on three different datasets.
- The authors perform a scaling experiment to benchmark the ability of their model to handle large datasets and show that it is scalable enough for typical contemporary large datasets.
- The ability of the model to learn explicit gene regulatory interactions is useful for interpretability and for learning the explicit gene networks governing differentiation.

**Weaknesses:**

- The authors perform an unsupervised classification task to predict the direction of regulation for a known TF-gene link from the TRRUST database. The paper does not benchmark the method's ability to predict the links themselves, e.g., by evaluating if the interactions with the highest predicted strength are enriched for known links. Furthermore, analysis of the top source or target genes using gene enrichment analysis could shed light into any relevant pathways.
- The authors mention GRN inference as a complementary line of work but only reference static GRN models. A few works have experimented with learning temporal gene networks for cell differentiation or biological state progression, such as Scenic+, Marlene, Dictys [1-3]. A discussion on how the current work differs from these or a direct comparison is needed to strengthen the contribution.
- The authors project data into 5 dimensions using PCA. This is too limiting for large single-cell datasets with ~20k genes and likely ignores finer interactions between genes. The authors need to present additional work/analysis that uncovers the type of information lost by doing such a projection and the model's ability to learn fine-grained interactions.

[1] https://www.nature.com/articles/s41592-023-01938-4
[2] https://academic.oup.com/bioinformatics/article/41/Supplement_1/i628/8199402
[3] https://www.nature.com/articles/s41592-023-01971-3

**Questions:**

Could the authors comment on the model's capability for extrapolation, that is, predicting a future time point beyond the final observed time in the training set?

---

> ### Author Response · Authors · 2025-11-20
> **Answer to Reviewer otWS**
>
> We thank the reviewer for their review and are happy about their generally positive reception of the proposed approach. In the following we will address their key concerns:
>
> # Answer to concerns
> - **Dynamic GRN inference methods.** *“...discussion on how the current work differs from these or a direct comparison is needed”*: We thank the reviewer for the very useful remark and believe that contrasting with respect to the temporal GRN inference methods is indeed important to fully capture the contribution of our method. We added a discussion to our related works section. Additionally, we provide two baselines (SCODE and Jacobian of OT-CFM) for the GRN discovery experiment and find that Cell-MNN outperforms both by 16 percentage points on average. Please refer to the general statement for details.
> - **Low dimensional PCA space.** *“...too limiting for large single-cell datasets with ~20k genes”*:
> This point has been shared by all reviewers and we think that it is very important and has not been addressed in enough detail in the paper. As it was a shared concern, we address it in the general statement.
> - **Predicting the existence of links.** *“...does not benchmark the method's ability to predict the links themselves”*: We agree with the reviewer that it would be highly desirable to also measure the performance of predicting the existence of regulatory links. As the TRRUST dataset does not resolve between *unknown* and *known-to-be absent* links, we found it hard to design a rigorous classification experiment to measure this performance. We now focused our time on the shared remarks by all reviewers (low dimensional PCA & missing GRN discovery baseline), however we would be happy to design an experiment for predicting the existence of links in the remainder of the discussion period, if the reviewer still thinks that this is an important experiment to perform (despite our point above).
>
> # Questions
> *“Could the authors comment on the model's capability for extrapolation, that is, predicting a future time point beyond the final observed time in the training set?”*
>
>  We have also asked ourselves this question and found the extrapolation problem to be significantly harder in practice. It goes beyond the task of interpolating between distributions and we find the performance of the model to be significantly worse at this task. We therefore do not advocate for the method to be used for extrapolation, but rather just interpolation.
>
> # Closing remarks
> We thank the reviewer for his feedback and hope that we were able to appropriately address the mentioned concerns. We remain available during the discussion period.

---

> > ### Author Response · Authors · 2025-12-01
> >
> > As a follow-up to our previous message, we would like to point the reviewer to the new experiment we performed to evaluate Cell-MNN's performance at predicting the *existence* of TF-gene links, as contained in TRRUST. We find that Cell-MNN performs competitively compared to all baselines (GRNBoost2, SCODE and OT-CFM) in terms of the AUROC score and Precision@K. We present a summary of the experiment in a new general message. All details of the experiment can be found in **Appendix E.1**.

---

### Official Review · Reviewer_51Jg · 2025-10-27

**Soundness:** 1
**Presentation:** 3
**Contribution:** 2
**Rating:** 2
**Confidence:** 5

**Summary:**

The paper proposes Cell-MNN, a ODE–based framework that learns explicit single-cell dynamics by modeling local linear ODEs in a PCA latent space. The approach avoids explicit OT preprocessing and claims to simultaneously provide interpretable gene regulatory interactions through the learned Jacobian matrices. Experiments on several benchmark single-cell datasets show competitive interpolation accuracy and advantages over OT-based baselines.

**Strengths:**

The paper is clearly written and tries to tackle an important problem in single-cell dynamic modeling — learning interpretable continuous-time dynamics.

**Weaknesses:**

1. Latent-space linearization is under-justified and fragile. The method hinges on representing dynamics as locally linear in a PCA latent space, $\dot z \approx A_\theta(z,t)z$, advanced via a matrix exponential. The paper itself acknowledges that evolving too far leaves the validity region and requires re-encoding to refresh the local ODE, but provides no analysis of linearization error growth, step-size control, or robustness to curvature in realistic trajectories. Moreover, the claimed efficiency advantage relies on a very low latent dimension; once d_z grows, computing and applying the operator (incl. eigendecomposition/inversion) becomes the dominant cost (see 4), which does not resolve the core limitations of Neural ODEs for stiff/strongly nonlinear flows— it only replaces numerical integration with piecewise-linear surrogates without guarantees.
2. GRN novelty is overstated; prior work is insufficiently acknowledged. Inferring gene regulatory relations from a learned ODE/vector field via its Jacobian has been proposed and used before (e.g., Dynamo [1]) and applied in multi-timepoint trajectory settings (e.g., TrajectoryNet [2], TIGON [3]). Cell-MNN can be viewed as the special case where the vector field is constrained to be piecewise linear in a low-dimensional PCA subspace. The “Gene Regulatory Network Discovery” related-work discussion omits or underplays these lines, which is misleading about the methodological novelty.
3. Scalability claims (vs. OT) are overstated, and the mechanism is unclear. The paper attributes scalability improvements to replacing OT preprocessing with MMD-based distribution matching and reports out-of-memory (OOM) errors for standard OT on inflated datasets. However, many Neural ODE–based or flow-matching methods already alleviate memory issues by computing OT or matching losses in mini-batches, thereby avoiding O(n^2) coupling matrices in practice. From this perspective, it is unclear why Cell-MNN “solves” sample-size scalability more effectively than these existing approaches, given that both rely on batched stochastic training. Moreover, substituting OT with balanced MMD does not directly address unbalanced dynamics (cell proliferation/death), where Unbalanced-SB frameworks provide principled formulations. Thus, the claimed scalability advantage seems more an artifact of implementation choices than a fundamental methodological improvement.
4. Cubic scaling in the latent dimension threatens practicality beyond very low $d_z$. The paper states $O(T d_z^2)$ for applying the analytical solution, but also a one-time $O(d_z^3)$ per operating point to form the operator (e.g., eigendecomposition/inversion). For $d_z=50–100$, this rapidly dominates and can exceed the cost of Neural ODE field evaluations, undermining the claimed efficiency. The assertion that 5D PCA is “sufficient” is not substantiated by ablations or biological coverage analyses. And I also believe this assertion is not correct.
5. Wall-clock efficiency is not compelling. Despite a 5D latent space and no OT, the reported runs surprisingly high (one hour) for the stated computational simplicity, and notably slower than many baselines under comparable settings. The paper does not explain where time is spent (e.g., repeated eigendecompositions, kernel MMD computation) or provide profiling.
6. Redundant/ill-motivated invertibility regularization. The additional term $L_{\text{inv}}(\theta)$, encouraging $P_\theta$ invertibility, appears unnecessary: invertible matrices are dense; a randomly initialized square matrix is almost surely invertible. The paper does not justify why this regularizer is needed, nor analyze its numerical side-effects (e.g., unstable gradients via $\det(\cdot)$).
7. Interpolation results of OT baselines are unclear. In Table 1, several OT-based methods underperform the OT-Interpolate baseline, while Cell-MNN surpasses it. Since many methods eventually regress toward OT-derived supervision, it is unclear why Cell-MNN outperforms those indirect OT-fitting approaches. The paper should articulate the mechanism (e.g., bias/variance advantages of distribution-level MMD vs. velocity-level supervision, regularization effects) and provide controlled comparisons
8. Amortized training procedure lacks essential details. Section 4.2 merges datasets in PCA space for amortized training and feeds a dataset index to the model, but does not explain the details.
9. Gene interaction validation lacks competitive baselines. Validation against TRRUST is useful, but the paper does not compare to other vector-field/Neural-ODE methods that could also extract Jacobian-based interactions under the same preprocessing. Without head-to-head comparisons, it remains unclear whether Cell-MNN offers any real advantage for GRN discovery beyond the convenience of an explicit linear operator.
10. Insufficient ablations/sensitivity analyses. Key design choices—kernel and bandwidth for MMD, discount factor $\gamma$, regularization weights $\lambda_{\text{kin}}$, $\lambda_{\text{inv}}$, latent dimension $d_z$, operator parameterization (e.g., fixing one eigenvalue to 0 vs. not ), and $\Delta t$ sampling—lack systematic ablations. Given that interpretability and stability hinge on these knobs, their impacts should be quantified.

**Questions:**

Due to these concerns, I lean towards rejecting the paper in its current form. Significant theoretical justification, experimental clarification, and comparison to prior ODE-based methods would be needed for a more favorable evaluation. The authors should:

1. Clarify justification for latent-space linearization. (See weakness 1)
2. Discuss the relation to prior GRN-from-ODE works. (See weakness 2)
3. Substantiate the scalability argument. (See weakness 3)
4. Provide complexity and latent-dimension analysis.  (See weakness 4)
5. Report detailed runtime and efficiency breakdown.(See weakness 5)
6. Revisit the invertibility regularization. (See weakness 6)
7. Clarify interpolation results and superiority to OT baselines. Could the authors explain why Cell-MNN achieves superior interpolation despite training toward similar objectives?  (See weakness 7)
8. Elaborate on amortized training implementation. (See weakness 8)
9. Enhance GRN validation comparisons (e.g., trajectorynet, TIGON). (See weakness 9)
10. Include hyperparameter and kernel sensitivity studies. (See weakness 10)

References
1. Xiaojie Qiu, Yan Zhang, Jorge D Martin-Rufino, Chen Weng, Shayan Hosseinzadeh, Dian Yang, Angela N Pogson, Marco Y Hein, Kyung Hoi Joseph Min, Li Wang, et al. Mapping transcriptomic vector fields of single cells. Cell, 185(4):690–711, 2022.
2. Alexander Tong, Jessie Huang, Guy Wolf, David Van Dijk, and Smita Krishnaswamy. Trajectorynet: A dynamic optimal transport network for modeling cellular dynamics. In International conference on machine learning, pages 9526–9536. PMLR, 2020.
3.  Yutong Sha, Yuchi Qiu, Peijie Zhou, and Qing Nie. Reconstructing growth and dynamic trajectories from single-cell transcriptomics data. Nature Machine Intelligence, 6(1):25– 39, 2024.

This review was independently written by the reviewer. An LLM was employed solely for minor phrasing and grammar improvements.

---

> ### Author Response · Authors · 2025-11-20
> **Answer #1 to Reviewer 51Jg**
>
> We thank the reviewer for his very detailed review and for the useful feedback. We are glad to hear that they found the paper *“...clearly written”* and that it *“..tackles an important problem”*. In the following, we address each of the reviewers' points and give additional empirical evidence where needed.
>
> # Answer to concerns
> 1) **Clarify justification for latent-space linearization.** *“...Latent-space linearization is under-justified and fragile.”*
>
> This points towards the core concept of the paper and we believe that there is an important methodological difference with respect to Neural ODEs that was potentially not stressed enough in the paper: We essentially propose to alter the learning task. Instead of learning a function that globally approximates the velocity field (map from state observation to time derivative), we propose to learn a function that maps from state observation to an interpretable function space of linear operators that themselves map from state observation to time derivative. The model thereby combines the expressivity of a Neural ODE with the interpretability of an explicit ODE. We empirically show that this approach performs competitively with the many methods proposed in the ML literature.
>
> On the remark that we might leave the validity region: We argue that the inferences in themselves are local. In the standard set-up of single-cell interpolation, the time point of the left-out marginal is closer to the input time than any of the training time points. There is actually no signal about more nonlinear dynamics between these time points.
>
> 2) **Discuss the relation to prior GRN-from-ODE works.** *“...GRN novelty is overstated”*
>
> We only present the GRN discovery experiment to show the secondary benefits of an explicit dynamics model and never claim that the GRN discovery is novel in itself. We adapted the related works section to highlight time-dependent GRN discovery methods such as Dynamo. For a more comprehensive comparison, we report the performance of two suitable baselines SCODE and Jacobian of OT-CFM. Cell-MNN outperforms both by an average of 16 percentage points. Please refer to the general statement for a more detailed answer to the shared remarks of all the reviewers about the GRN discovery experiment.
>
> 3) **Substantiate the scalability argument.**
>
> We agree with the reviewer pointing out that *“...many Neural ODE–based or flow-matching methods already alleviate memory issues by computing OT or matching losses in mini-batches”*. This is true, however, they do so with strong heuristic approximations. Using OT to precompute the matchings in itself is already a hard and debatable assumption (see point number 8) about the underlying dynamics and approximating it on minibatches is even more limiting. Cell-MNN in contrast, does adaptive matching as part of the learning process while being scalable to larger datasets and outperforming the minibatch-OT baseline.
>
> 4) **Provide complexity and latent-dimension analysis.** *“...The assertion that 5D PCA is “sufficient” is not substantiated”*
>
> As this concern was shared by multiple reviewers, we address it in the general statement.
>
> 5) **Report detailed runtime and efficiency breakdown.** *“...Despite a 5D latent space and no OT, the reported runs surprisingly high (one hour)”*
>
> We adaptively learn the coupling that for other methods is precomputed using the OT preprocessing. This means that we have to learn significantly more during training compared to methods that rely on couplings obtained by OT. We would like to emphasize that we do not claim faster performance on small datasets but rather beneficial scaling to larger sample sizes where global OT methods run into out-of-memory errors. We are happy to report profiling results during the remainder of the discussion period.
>
> 6) **Revisit the invertibility regularization.** *“...additional term , encouraging  invertibility, appears unnecessary: invertible matrices are dense”*
>
> We remark that invertible matrices are not necessarily dense (e.g. diagonal matrices). Furthermore, encouraging non-zero determinant on the operators does not force them to be dense. We introduce the regularizer into the loss to ensure that the operators predicted by the encoder stay invertible throughout training. This is crucial to keep the training process stable, as it requires inversion of the matrix in the training loop. We run nine example training runs on the Embryoid dataset without the regularizer and incur training instabilities for 4/9 of them (training loss suddenly jumping up and not recovering).

---

> > ### Author Response · Authors · 2025-11-20
> > **Answer #2 to Reviewer 51Jg**
> >
> > 7) **Clarify interpolation results and superiority to OT baselines.** *“...it is unclear why Cell-MNN outperforms those indirect OT-fitting approaches.”*
> >
> > There is no a priori reason why OT should be the ground truth process. We argue that it is a useful but limited modeling assumption. This is supported by our empirical result (Table 1) that adaptive learning of Cell-MNN (that is more agnostic towards the underlying dynamics) leads to better predictive performance. For OT-Interpolate we simply compute the OT map between consecutive time points and linearly interpolate this map to infer the marginal distribution at the left out time point. These are in a way the data that methods based on OT preprocessing treat as the ground truth used for training, which explains the strong performance. However the more agnostic approach of Cell-MNN shows improved empirical performance, supporting our point above.
> >
> > 8) **Elaborate on amortized training implementation.** *“...Amortized training procedure lacks essential details”*
> >
> > The description of the amortized training set-up is given in *Section 4.2 (line 355-361)*. In case this is still unclear, we kindly ask the reviewer to specify which details they find to be missing.
> >
> > 9) **Enhance GRN validation comparison.** *“...does not compare to other vector-field/Neural-ODE methods”*
> >
> > We provide SCODE and the Jacobian of OT-CFM as two suitable baselines and show that Cell-MNN outperforms both by 16 percentage points. We give a detailed response to the remarks on the GRN experiment in our general statement.
> >
> > 10) **Include hyperparameter and kernel sensitivity studies.** *“Insufficient ablations/sensitivity analyses.”*
> >
> > We are currently working on the ablation studies on all hyperparameters mentioned by the reviewer and will report the results as soon as they are available.

---

> > > ### Comment · Reviewer_51Jg · 2025-11-21
> > >
> > > I sincerely appreciate the authors’ detailed response. The paper is now clearer. With the provided clarifications and new experiments, I believe the paper has been greatly strengthened, and several of my earlier concerns have been addressed. I believe this work has merits, so I am willing to raise my score accordingly. However, some issues remain insufficiently convincing, and I still hold certain reservations. While I appreciate the additional biological analysis involving 5D PCA, my primary concern lies in the method’s scalability and computational efficiency. It would greatly strengthen the paper if the authors could demonstrate the scalability of the method in higher-dimensional spaces (e.g., 50-dimensional and 100-dimensional), and report the corresponding runtime and memory usage. A comparison of these metrics against other relevant baselines would further convince me of the practical effectiveness of the proposed approach.

---

> > > > ### Author Response · Authors · 2025-11-26
> > > >
> > > > We thank the reviewer for their fast reply and are glad to hear that our response helped clarify some of the points and concerns of the reviewer. To address the reviewers' concerns on scalability and computational efficiency, we perform the experiment suggested by the reviewer and train in 50 dimensional and 100 dimensional PCA subspace. Please refer to the general message for a summary of the results and to **Appendix F.1** for all the details. We think that this was an insightful experiment to perform and thank the reviewer for the suggestion.

---

> > > > > ### Comment · Reviewer_51Jg · 2025-11-26
> > > > >
> > > > > Thank you. I am curious about the added results. Table 3 reports that running the method in 5D takes approximately one hour. Yet in the new experiments using 50- and 100-dimensional PCA spaces, the reported runtime is 4–30 minutes, which appears to be a substantial speed-up despite the much higher dimensionality. Could the authors clarify why increasing the dimensionality results in faster computation?

---

> > > > > > ### Author Response · Authors · 2025-11-26
> > > > > >
> > > > > > Thank you for your quick reply. The speed-up in run time is due to two changes we make in the high dimensional experiment: firstly, we move to a GPU with more RAM and use a larger batch size, which speeds up convergence. Secondly, we set the early stopping patience from 40 to 10 evaluation checks, which also speeds up training significantly (probably with some expense in performance).
> > > > > >
> > > > > > We note that we did not treat runtime as an important design decision in previous experiments, but thought of ways to improve it due to the time constraint of the discussion period and the concerns of the reviewer (which were more focused on practicality and less on pure performance).

---

### Official Review · Reviewer_4yFL · 2025-10-31

**Soundness:** 3
**Presentation:** 3
**Contribution:** 2
**Rating:** 2
**Confidence:** 4

**Summary:**

This paper introduces a framework for learning a locally linear differential equation in latent space that describes gene expression changes during cell differentiation. The latent space itself is also a linear projection from gene space. A key idea of the paper is to make the dynamics analytically tractable so that gene interactions can be explicitly modeled.

**Strengths:**

Strengths:
•	The idea is well-motivated, theoretically sound, and biologically meaningful.
•	The idea is clearly explained. The paper is well-written and I found it easy to read
•	The evaluations presented are overall sound and sensible.

**Weaknesses:**

Weaknesses:
•	Highly similar concept to VeloVAE, ICML 2022. VeloVAE fits a mixture of linear ODEs and uses an encoder-decoder framework to perform Bayesian inference of cell times and ODE parameters. Another highly similar paper is LatentVelo, which fits a neural ODE end-to-end to learn dynamics in the latent space. Dynamo uses a linear ODE to estimate the Jacobian of gene regulation.
•	The method seems more incremental than revolutionary. There are closely related approaches for the same problem. The move to a locally linear ODE doesn't seem that impactful to me.
•	Baseline chosen for gene regulatory network evaluation is very weak. There are dozens of gene regulatory network inference algorithms that take single-cell RNA-seq data as input. Comparing performance against these would be more informative. Dynamo (Qiu et al. Cell 2022) seems particularly relevant, because a stated goal of the method is to recover gene-gene interactions by estimating the Jacobian.
•	I understand the motivation for interpretability, but it seems that this local linearity would have to come with some loss of predictive power. Approaches like VeloVAE don’t suffer from this limitation while retaining interpretability. I don't understand how this restricted model can outperform less interpretable but more expressive models, apart from scalability concerns.
•	Evaluation in terms of rate parameters seems important. The locally linear ODE can be interpreted in terms of gene expression changes (RNA velocity), and thus it's important to benchmark against the class of methods that aim to estimate these rates directly. Ground truth in the form of metabolic labeling data (see Dynamo paper) is available for an increasing number of datasets.

**Questions:**

1. How is your approach different from VeloVAE and LatentVelo?
2. Why does your less expressive approach outperform more expressive previous models? This seems surprising, because I would expect the local linearity constraint to make the predictions less accurate.

---

> ### Author Response · Authors · 2025-11-20
> **Answer to Reviewer 4yFL**
>
> We thank the reviewer for their very useful feedback and are happy to hear that they found the idea *“...well-motivated, theoretically sound”* and the evaluations overall *“...sound and sensible”*. We address the reviewer concerns and questions below and remain available for any additional clarification during the discussion period:
>
> # Concern: Comparison to other methods
> The reviewer raised the concern that Cell-MNN uses a *“Highly similar concept to VeloVAE”* and that a *“...highly similar paper is LatentVelo”* and that there are *“...closely related approaches for the same problem”*. In our view, this concern stems from conflating methods that operate on fundamentally different types of input data. In the following, we summarize the differences with respect to input data modalities and algorithmic details.
> - **Different input regimes in scRNA-seq.** In the context of scRNA-seq there are three relevant data regimes: (i) *spliced* and *unspliced* RNA counts, (ii) *metabolically labeled* RNA counts, and (iii) “plain” *UMI gene expression* counts. The first two provide two quantities per gene with a known temporal ordering, which can be used to approximate a local RNA velocity vector. In contrast, pure UMI counts only provide a single expression level per gene per cell.
> - **Velocity-based methods solve a different problem.** Dynamo, VeloVAE, and LatentVelo are explicitly designed for settings with spliced or metabolically labeled RNA, and model or use RNA velocity fields as input. Their goal is to reconstruct dynamics from velocity estimates derived from these richer measurements. By construction, they cannot be applied to datasets that only provide pure UMI counts without splicing or labeling information.
> - **Algorithmic differences:** In addition to the different data modalities, VeloVAE and LatentVelo are algorithmically different from Cell-MNN: Both parametrize their ODE specifically for splicing dynamics, which as described above, is a different problem.
> - **Cell-MNN operates on pure UMI counts.** Cell-MNN, and the machine learning baselines we compare with, take only gene expression counts (UMIs) together with sampling times as input. This is the data modality produced by standard “plain” scRNA-seq protocols (e.g., 10x Genomics) and by the established single-cell interpolation benchmarks we evaluate on.
> - **Consequences for comparability.** Because they assume different input modalities and model different objects, velocity-based methods such as Dynamo, VeloVAE and LatentVelo are not suitable for comparison. Instead, we benchmark with SOTA methods that operate on the same input type and task as Cell-MNN **(see the nine methods we compare with in Table 1)**.
> - **Advantages of working on pure UMI data.** Focusing on pure UMI counts makes these methods broadly applicable: any time-resolved scRNA-seq dataset with gene expression counts can be used, without requiring specialized splicing or metabolic labeling protocols. This is particularly important for amortized training across multiple datasets with varying experimental designs.
> - **Adjustment of paper:** If the reviewer thinks that the above described distinction is relevant for future readers, also to describe the conceptual difference with respect to VeloVAE and LatentVelo, we would be happy to incorporate it into the related works section of the paper.
>
> # Concern: *“Baseline chosen for gene regulatory network evaluation is very weak”*
> We added SCODE and Jacobian of OT-CFM to our GRN discovery experiment and show that Cell-MNN outperforms both. Please refer to the general statement for a detailed response to the shared concerns on the GRN discovery experiment.
>
> # Questions
> *Why does your less expressive approach outperform more expressive previous models?*
>
> Cell-MNN is actually as expressive as any black box Neural ODE as long as the dynamics stay in the approximation region of the local linear dynamics and the condition $f(z=0,t)=0\, \forall \, t$ is fulfilled. Moreover, the linear operators are a highly non-linear function of the input, which is where the expressivity lies. For intuition: As any arbitrary smooth function can be locally linearized, Cell-MNN can fit any such function by predicting that local linearization. The empirical performance supports the idea that the gene expression trajectories are smooth enough, such that the regions of approximation are large enough to stretch to the next time point of the dataset.
>
> # Closing remarks
> We thank the reviewer for their valuable feedback and hope that the reviewer's concerns were precisely addressed. We remain available for additional questions or remarks during the discussion period.

---

> > ### Comment · Reviewer_4yFL · 2025-11-26
> >
> > Thank you for your thoughtful responses. I would like to follow up on two points.
> > 1. I appreciate that RNA velocity approaches take spliced and unspliced counts as input, whereas your approach takes only UMI counts as input. However, both types of inputs are derived from the same biological experiments (single-cell RNA-seq), just with different preprocessing steps. In fact, the requirement for multiple time points means that RNA velocity methods are more broadly applicable, because most single-cell RNA-seq datasets have only a single timepoint. Additionally, the interpretation of your gene-space parameters is very similar to the notion of RNA velocity, and thus the goals of your modeling are very similar.  Do you have any responses to these points?
> > 2. You provided an explanation for why the linearization does not significantly reduce expressivity, but why do you think your approach actually outperforms previous models that do not rely on this linearization?

---

> > > ### Author Response · Authors · 2025-11-27
> > >
> > > Thank you for your answer and for describing where you still find the paper unclear. We will address your two points in the following:
> > > 1. Apart from the technical detail of different input data, RNA velocity methods do indeed also aim at solving a different biological problem: They look at the lifecycle of mRNA molecules and derive *local* velocity estimates from it. This analysis is effective at significantly *smaller* timescales (a few hours as opposed to one or more days) and, as the reviewer mentioned, thereby works within a single time point. On the other hand, approaches such as Cell-MNN that connect multiple timepoint measurements aim at the analysis of *longer, multi-step* processes such as organ development, reprogramming, or trans-differentiation. They thereby also require multiple time point measurements. In summary, the distinction can be condensed to whether a methodology aims at modeling gene expression changes on time scales on the order of mRNA lifetimes (therefore short-term, usually a couple of hours, and for the datasets we use, the effect would be only within-timepoint), as opposed to cell fate changes that require longer time scales. We hope that this explanation clarifies that the goals of these methods are distinct. We remark that fundamentally different approaches could potentially be merged in a synergistic way, however this is not the aim of our paper, and we are happy to mention this as possible future work.
> > > 2. The fact that the linearization of dynamics between time points leads to better interpolation performance is likely due to the dynamics being indeed linear in this regime. The inductive bias thereby effectively constrains the search space of solutions and leads to improved performance. Note that the data in itself also does not directly contain information about potential non-linear dynamics in-between the grid of observed time points. Our results seem to indicate that constraining the model in this way fits well with the structure of the data at hand.

---

### Author Response · Authors · 2025-11-20
**General Answer**

We appreciate the detailed reviews and would like to thank all reviewers for their feedback and time invested. We are happy to hear that the reviewers found the paper to be *“...well-motivated, theoretically sound”* (4yFL), *“...clearly written”* (51Jg) and that it tackles *“...an important problem”* (51Jg). There are two clear concerns that are shared by multiple reviewers and we address them in this general statement. Questions or remarks that are particular to individual reviewers are addressed in direct answers.

# Concern of all reviewers: Low dimensional PCA space
All the reviewers raised concern about the use of a low dimensional PCA space to model the gene expression dynamics in. We believe these concerns arise from a misunderstanding of the scope of our contribution:
- **Low-dimensional PCA is standard in ML-based trajectory inference.** Using 5 principal components is common practice in the ML literature on single-cell interpolation, **see the nine methods we compare with (see Table 1 in the paper)**. This aligns with the assumption in computational biology that scRNA-seq data lie on a low-dimensional manifold. For example, in \[1] trajectory inference is performed directly on 5 PCs to derive biological conclusions about lineage progression.
- **Why low-dimensional PCA is biologically meaningful.** Across all datasets we evaluate on, PCA variance plots show that the first few components capture the large part of the variance, see Figure 7. Using 5 PCs yields a cumulative explained variance of above 60% for Cite and Multi and above 40% for the Embryoid dataset. Crucially, the 5 dimensional PCA embedding preserves cell-type information: We compute the KNN classifications performance using 15 neighbors and observe an accuracy of 87 % (Multi) and 90% (Cite) (Embryoid has no cell-type labels). To visualize this, we show the UMAPs computed on the 5 dimensional PCA embedding (Figure 7b & d) clearly cluster by cell-type. This is the essential requirement in scientific studies that investigate lineage bifurcations, where the goal is to predict cell-type transitions and not reconstruct all gene-level nuances.
- **Our contribution within this setting.** Our paper is **not** about learning a different representation for this problem. We propose Cell-MNN to improve the learning algorithms on top of the commonplace representations used in other deep learning baselines. Cell-MNN achieves SOTA average performance while avoiding strong OT assumptions and providing explicit, local dynamics that are interpretable as gene interactions.
- **Clarifications in the revision.** We thank the reviewers for pointing out that the low-dimensional PCA embedding was not sufficiently discussed. We added a dedicated section (Appendix C.1)  including (i) motivation (ii) PCA variance plots for all datasets, (iii) a demonstration that cell-type structure is preserved in 5-PC space using KNN classification and UMAP plots, and (iv) comparison of OT-CFM and Cell-MNN with 10-PCs (similar performance).

\[1]: https://www.nature.com/articles/s41467-019-11493-2

# Concern of all reviewers: GRN discovery baseline is too weak
All reviewers were asking for additional baselines for the GRN discovery experiment. We completely agree that additional baselines make the picture more comprehensive and thank the reviewers for the suggested new baselines. We have updated our GRN discovery experiment showing outperformance with respect to SCODE and the Jacobian of OT-CFM. In the following we present our line of reasoning:
- **Requirement: prediction of signed gene interactions.** Our GRN experiment evaluates whether Cell-MNN can correctly classify regulatory interactions as *activating* or *repressing*. This focuses on signed edges, which substantially restricts the set of meaningful baselines.
- **Many popular GRN methods do not predict signed edges.** Methods such as Marlene, GRNBoost, and GENIE3 output only unsigned regulatory edges. This means that they cannot serve as baselines for our task.
- **Velocity-based methods require incompatible data.** Dynamo and related RNA-velocity-based GRN estimators depend on spliced or metabolically labeled RNA counts. The datasets in our study provide only pure UMI counts, making these methods inapplicable.
- **GRN methods based on scATAC-seq are not applicable.** Methods such as SCENIC+ and Dictys require paired chromatin accessibility (scATAC-seq) and scRNA-seq measurements. These are not available for the datasets used in our study, and hence these methods cannot serve as baselines.
- **Added competitive baselines.** To strengthen the evaluation we have updated the GRN discovery experiment with additional baselines that do output signed interaction matrices: (i) the Jacobian of OT-CFM, and (ii) SCODE. As reported in Table 2, Cell-MNN outperforms the two baselines by around 16% percentage points on average, supporting the claim that the predicted operators capture biologically meaningful interactions.

---

> ### Author Response · Authors · 2025-11-26
> **High dimensional experiments (50 and 100 dimensions) as follow up to 5D PCA concerns**
>
> To investigate the applicability of Cell-MNN to higher dimensional latent spaces, we train it in 50 dimensional and 100 dimensional PCA subspaces. We only adapt the learning rate and early-stopping patience and observe training times between 4m 25s and 31m 29s, with GPU RAM usage below 25GB. Without additional hyperparameter tuning, Cell-MNN performs within error bars of the SOTA method for the Multi dataset and remains competitive with other methods on the Cite dataset. As described in our previous response, we remark that working with such high dimensional PCA subspaces goes beyond typical workflows in trajectory inference. However still, we think that these results are meaningful in investigating the practical applicability of Cell-MNN to higher dimensional latent spaces and that, perhaps, we have been too conservative in Section 2.1 (Limitations) concerning the scalability challenges of our method. Thank you for suggesting us to investigate this. A detailed experiment description and the results are provided in **Appendix F.1**.

---

### Author Response · Authors · 2025-12-01
**General Answer**

# Concern: Evaluation of link-existence prediction
Some of the reviewers remarked that the paper *“... does not benchmark the method's ability to predict the links themselves”* (otWS) and that they would have liked to see *“...comparison to methods like GRNBoost or Genie3 on edge existence”* (JgxX).

To address this, we designed a new experiment to measure Cell-MNN's performance at predicting the *existence* of regulatory interactions. We compare with the commonly used GRN discovery methods GRNBoost2 and SCODE and find that Cell-MNN performs competitively compared to all baselines in AUROC score and the Precision@K metric, indicating that it is indirectly learning to predict the existence of interactions (we find GENIE3 to be intractable for the problem, due to the number of potential TF-gene links).

**Experiment Summary.** We train all models on the Embryoid dataset (Moon et al. 2019) and measure performance with respect to links listed in the TRRUST database. To restrict TRRUST to interactions that are plausibly involved in the differentiation dynamics of the Embryoid dataset, we subset the database to interactions whose transcription factor regulator is mentioned as relevant in the original analysis of Moon et al. (Fig. 6d). All details on the experiment can be found in **Appendix E.1**.

---

### Author Response · Authors · 2025-12-01
**Message to the Area Chair**

Dear Area Chair,

We highly appreciate the engaging discussions with all reviewers. Their feedback and suggestions significantly improved our paper. The push towards more comprehensive evaluation by Reviewers 51Jg and JgxX led to insightful new results. In particular Reviewer 51Jg (*confidence 5*) had two primary concerns: first, the inclusion of competitive baselines for our GRN discovery experiment, after which they increased their score to 4; and second, the evaluation of performance in higher-dimensional latent spaces, which led them to further raise their score to 6.


To assist the Area Chair in their evaluation, we summarize the concerns of the reviewers and our rebuttal during the discussion phase in the following bullet points.

- **Low dimensional PCA space:** All reviewers raised concern about the use of a low dimensional PCA subspace to model the dynamics. In our rebuttal, we showed that 5 dimensional PCA is biologically meaningful by quantitatively verifying that the embeddings contain cell-type information, which is essential for lineage bifurcation analysis **(Appendix D.1)**. We furthermore added an experiment for 50 and 100 dimensional PCA space, finding that Cell-MNN performs within error bars of the SOTA method on the Multi dataset while remaining competitive on the Cite dataset **(Appendix F.1)**. This analysis led reviewer 51Jg to raise their score twice.
- **GRN baseline:** All reviewers remarked that the experiment of predicting interaction types (*activating* or *repressing*) was missing strong baselines. To address this concern, we added two baselines (SCODE and the Jacobian of OT-CFM) to our original GRN evaluation experiment. We found that Cell-MNN outperforms both **(Table 2)**.
- **GRN link existence prediction:** Reviewers JgxX and otWS suggested additionally evaluating Cell-MNN’s performance at predicting the *existence* of regulatory links. In our rebuttal, we evaluated Cell-MNN and three baselines (GRNBoost2, SCODE, and the Jacobian of OT-CFM) on predicting the existence of regulatory links listed in TRRUST. We found that Cell-MNN performs competitively compared to these three GRN discovery baselines **(Appendix E.1)**.
- **Comparison with RNA velocity methods:** Reviewer 4yFL raised concerns about the novelty of our approach and asked how it differs from RNA velocity methods. In the discussion phase, we clarified that RNA velocity methods (including those mentioned by the reviewer) address a different biological problem, modeling dynamics at a different time scale. Please refer to our detailed response to Reviewer 4yFL for further details.

We hope that outlining the discussion helps the Area Chair in following the different strings of the conversation of the discussion period. We look forward to hearing about the Area Chairs assessment of the paper and thank them for their effort invested.

Thank you for your service to the ICLR community.

Kind regards,

The Authors

---

### Meta-Review · Area_Chair_eq6v · 2026-01-18

**Summary:**

This paper is at the margin of acceptance. No reviewer strongly championed this paper, and many concerns were raised in the initial reviews. After rebuttal, a significant number of concerns were addressed, and an initially negative reviewer indicated willingness to raise their score. Meanwhile, the major concern from Reviewer 4yFL is partially clarified.

From my perspective, the method proposed is interesting, and the problem it tackles is important -- and thus I recommend acceptance of the paper.

**Reviewer Concerns:**

Most technical concerns, e.g., the low-dimensional PCA justification and scalability (including 50/100D experiments with computation report), strengthened GRN baselines, and the added evaluation -- were addressed in the rebuttal. However, the “incremental contribution” concern raised by Reviewer 4yFL is not fully resolved.

**Reviewer Scores:**

4yFL (originally 2): Keep at 2 or might raise to 4. The rebuttal might clarify the incremental nature of the paper a bit, but it might be hard to reverse the negative rating.

51Jg (originally 2): Likely raised to 6. Most concerns are solved.

otWS (originally 6): Likely unchanged. Major concerns are solved.

JgxX (originally 6): Likely unchanged. Major concerns are solved.

---

### Decision · Program_Chairs · 2026-01-26

Accept (Poster)